# SureMap: Simultaneous mean estimation for single-task and multi-task disaggregated evaluation

**Mikhail Khodak**\*
Princeton University
mkhodak@cs.cmu.edu

**Lester Mackey, Alexandra Chouldechova, Miroslav Dudík**
Microsoft Research
{lmackey,alexandrac,mdudik}@microsoft.com

## Abstract

Disaggregated evaluation—estimation of performance of a machine learning model on different subpopulations—is a core task when assessing performance and group-fairness of AI systems. A key challenge is that evaluation data is scarce, and subpopulations arising from intersections of attributes (e.g., race, sex, age) are often tiny. Today, it is common for multiple clients to procure the same AI model from a model developer, and the task of disaggregated evaluation is faced by each customer individually. This gives rise to what we call the *multi-task disaggregated evaluation problem*, wherein multiple clients seek to conduct a disaggregated evaluation of a given model in their own data setting (task). In this work we develop a disaggregated evaluation method called **SureMap** that has high estimation accuracy for both multi-task *and* single-task disaggregated evaluations of blackbox models. SureMap's efficiency gains come from (1) transforming the problem into structured simultaneous Gaussian mean estimation and (2) incorporating external data, e.g., from the AI system creator or from their other clients. Our method combines *maximum a posteriori* (MAP) estimation using a well-chosen prior together with cross-validation-free tuning via Stein's unbiased risk estimate (SURE). We evaluate SureMap on disaggregated evaluation tasks in multiple domains, observing significant accuracy improvements over several strong competitors.

## 1 Introduction

Evaluation is a key challenge in modern AI, with much effort spent deciding what metrics to measure, with which methods, and on what data. This challenge is especially acute in fairness assessment, which requires not only high-quality data to run a model and score its outputs but also demographic information for defining groups. Due to the high cost of obtaining high-quality evaluation data, the issue of sample complexity—sample size needed to get a good performance estimate—remains salient, especially when we want to release not just one overall measure but instead to output a *disaggregated evaluation* that captures variation among demographic subpopulations of the data [Barocas et al., 2021]. For instance, we might want to assess group fairness by examining the variation in performance across groups of users defined by intersections of the demographic attributes age, race, and sex. The naive approach of independently evaluating each group's performance on its own data can fail because the sample sizes of intersectional groups rapidly decrease as we consider more attributes [Herlihy et al., 2024]. Recent work has shown how to improve upon naive methods by combining data from multiple subpopulations to inform their individual performance estimates [Miller et al., 2021, Herlihy et al., 2024].

In today's technology landscape it is common for multiple clients to *procure* the same model (e.g., an automated speech recognition or language model) from an AI developer, with each client performing a disaggregated evaluation of *the same model* on *their own data*. We refer to this problem as the **multi-task disaggregated evaluation**. We formalize and study this problem, showing that one can improve the disaggregated evaluations of individual clients by using multi-task data in the form of

---

\*Work done in part while at Microsoft Research and supported in part by a CMU TCS Presidential Fellowship.

38th Conference on Neural Information Processing Systems (NeurIPS 2024).

summary statistics from other clients or from the model provider. Our approach uses the (out-of-distribution) multi-task data to set the parameters of a multivariate normal prior and then performs *maximum a posteriori* (MAP) inference on the (in-distribution) client data. Formally, we model the problem as Gaussian mean estimation and design a simple-yet-expressive additive prior that can capture many different relationships between subpopulations. Drawing upon classical statistics, we fit prior parameters by minimizing Stein's unbiased risk estimator [SURE, Stein, 1981]. While motivated by multi-task considerations, we show that our method also performs well in the single-task setting.

## 1.1 Contributions

1. **SureMap**: We introduce a method that uses SURE to tune the parameters of a well-chosen Gaussian prior before applying MAP estimation. The prior is motivated by its attainment of a good efficiency–expressivity tradeoff, requiring only a linear (in the number of subpopulations) number of parameters to recover several natural baselines for disaggregated evaluation.
2. **Datasets**: Disaggregated evaluation has few benchmarks [Herlihy et al., 2024], so we introduce new ones for both the single-task and multi-task settings, covering automated speech recognition (ASR) and also tabular domains (with linear models and also in-context LLMs).
3. **Single-task:** We find that SureMap is always competitive with strong baselines from prior work, while improving significantly in some settings with intersectional sensitive attributes.
4. **Multi-task:** Incorporating data from multiple clients into SureMap yields significant improvements across all evaluated settings. This multi-task approach is more accurate even with just one additional task and is the only method to consistently outperform the naive and pooling baselines.

## 1.2 Related work

Disaggregated evaluation is a core task in the fairness assessment of AI systems [Barocas et al., 2021]. Past work has sought to improve estimation accuracy by combining information across different groups, e.g., via Bayesian modeling [Miller et al., 2021], Gaussian process approximation of loss surfaces [Piratla et al., 2021], and structured regression [Herlihy et al., 2024]. The last work found that classical James–Stein-type mean estimation [James and Stein, 1961, Bock, 1975] is often competitive, and so we adopt it as our first non-naive baseline. We also compare to structured regression itself, which turns out to have a tight mathematical connection to SureMap; indeed, apart from our use of Gaussian (ridge) rather than Laplace (lasso) priors (regularization)—as well as our use of a more flexible tuning based on SURE rather than cross-validation—the method of Herlihy et al. [2024] can be viewed as the discriminative counterpart to our generative approach (see §E for details).

Within the disaggregated evaluation literature we are the first to formulate and study *multi-task* disaggregated evaluation. This is an important direction because (a) model providers often have their own data or data from multiple clients that can inform the evaluation and (b) transferring information across distributions is a key way to handle very low-sample regimes. We also contribute several datasets that we hope will spur further development in disaggregated evaluation.

SureMap relies on applying classical mean estimation tools to quantities modeled as Gaussian means. Notably, Miller et al. [2021] model *scores* via well-studied distributions—e.g., Gaussians—but since scores are related non-linearly to metrics it is unclear if this can lead to similarly simple estimators. To tune parameters, we use SURE, a popular statistical approach [Li, 1985, Donoho and Johnstone, 1995]. Specifically, in the empirical Bayes tradition, we use it to set the MAP estimator of a hierarchical model. Using SURE to tune the scale of an isotropic Gaussian prior was shown to be asymptotically (in the dimension) optimal in the case of heteroskedastic data distributions [Xie et al., 2012]. Since disaggregated evaluation data is highly heteroskedastic due to variation in group size, this is positive evidence for our approach, although our prior is non-isotropic and has many more variance parameters.

## 2 Setup

We first describe the disaggregated evaluation problem (§2.1), recast it as a Gaussian mean estimation (§2.2), and motivate a multi-task variant (§2.3), all while introducing several baselines estimators.

## 2.1 Setting and baselines

We want to assess a predictive model $p : \mathcal{X} \to \mathcal{Y}$ under some distribution $\mathcal{D}$ over input space $\mathcal{X}$ and output space $\mathcal{Y}$ using error measure $\ell : \mathcal{Y} \times \mathcal{Y} \to \mathbb{R}$. For example, in image classification, $\mathcal{D}$ is a distribution over (image, label) pairs and $\ell$ is the 0-1 error. To simplify notation, we mainly deal with the composite function $f(z) = \ell(y, p(x))$ acting on points $z = (x, y)$ in the product space $\mathcal{Z} = \mathcal{X} \times \mathcal{Y}$.

In *disaggregated* evaluation, $\mathcal{Z}$ is assumed to be a union of $d \geq 1$ disjoint subsets $\mathcal{Z}_g$, each associated to some subpopulation or *group* $g \in [d]$, where we use $[k]$ to denote the set $\{1, \ldots, k\}$. As a running example, suppose each point $z \in \mathcal{Z}$ is an individual whose sex $s$ and age $a$ are categorical variables with $d_1$ and $d_2$ possible values, respectively. Then $d = d_1 d_2$ and each point has an associated index $g = (s - 1)d_2 + a$ denoting its intersection of sex $s \in [d_1]$ and age $a \in [d_2]$. The task is then to use a set $S \sim \mathcal{D}^n$ of $n \geq 1$ i.i.d. samples from the distribution $\mathcal{D}$ to estimate the vector of true subpopulation errors $\boldsymbol{\mu} \in \mathbb{R}^d$, with components $\mu_g = \mathbb{E}_{z \sim \mathcal{D} \mid \mathcal{Z}_g}[f(z)]$.[1] We write $\hat{\boldsymbol{\mu}}(S) \in \mathbb{R}^d$ for an estimator of $\boldsymbol{\mu}$ with components denoted as $\hat{\mu}_g(S)$.

Empirically, we measure estimation accuracy, i.e., the performance of the performance estimate, via **mean absolute error** (MAE) $L_{\boldsymbol{\mu}}^{\text{MAE}}(\hat{\boldsymbol{\mu}}(S)) = \frac{1}{d}\|\hat{\boldsymbol{\mu}}(S) - \boldsymbol{\mu}\|_1$, which is easy to interpret and less sensitive to outliers than **mean squared error** (MSE). Our method development, however, is based on a count-weighted version of MSE, where we denote group counts as $n_g = |S \cap \mathcal{Z}_g|$:

$$L_{\boldsymbol{\mu}}^{\mathbf{n}}(\hat{\boldsymbol{\mu}}(S)) = \frac{1}{d} \sum_{g=1}^{d} n_g(\hat{\mu}_g(S) - \mu_g)^2. \tag{1}$$

We conclude the setup with two baselines. The first is the **naive estimator** returning group means:[2]

$$\hat{\mu}_g^{\text{naive}}(S) = \frac{1}{n_g} \sum_{z \in S \cap \mathcal{Z}_g} f(z). \tag{2}$$

While unbiased, $\hat{\boldsymbol{\mu}}^{\text{naive}}$ can perform poorly on groups with few samples. The second baseline, the **pooled estimator**, returns an identical quantity—the overall sample mean—for all groups:

$$\hat{\mu}_g^{\text{pooled}}(S) = \frac{1}{n} \sum_{z \in S} f(z) = \frac{1}{n} \sum_{g=1}^{d} n_g \hat{\mu}_g^{\text{naive}}(S). \tag{3}$$

This estimator is generally biased (unless all group means are equal), but it can perform well in low-sample regimes thanks to a much lower variance.

## 2.2 A Gaussian model for disaggregated evaluation

We use a simple but natural model to aid in the design of disaggregated evaluation methods. Specifically, denoting the naive estimator by $\mathbf{y} = \hat{\boldsymbol{\mu}}^{\text{naive}}(S) \in \mathbb{R}^d$, we model group $g$'s entry $y_g$ as being drawn from a Gaussian with (unknown) mean $\mu_g$ and (known) variance $\sigma^2/n_g$, where $\sigma^2$ is shared across groups. This reduces the problem of disaggregated evaluation, as defined in §2.1, to that of estimating the mean of a multivariate Gaussian with known diagonal covariance $\Sigma_{g,g} = \sigma^2/n_g$ given a single sample $\mathbf{y} \sim \mathcal{N}(\boldsymbol{\mu}, \boldsymbol{\Sigma})$. Our model has many advantages in the disaggregated evaluation setting:

1. By the central limit theorem, $\mathbf{y}$ is asymptotically normal with mean $\boldsymbol{\mu}$ and diagonal covariance $\boldsymbol{\Sigma}$ for many distributions $\mathcal{D}$ of interest, even when the underlying data is non-Gaussian. Furthermore, because the methods derived from our model only take $\mathbf{y}$ and $\boldsymbol{\Sigma}$ as input, they can be applied even when the evaluated statistic is not the pointwise average assumed by the setup in §2.1, so long as $\mathbf{y} \sim \mathcal{N}(\boldsymbol{\mu}, \boldsymbol{\Sigma})$ holds asymptotically. An example of this is when $y_g = \hat{\mu}_g^{\text{naive}}(S)$ corresponds to the area under the ROC curve (AUC) computed over group $g$'s data $S \cap \mathcal{Z}_g$ [Lehmann, 1951]; we demonstrate SureMap's applicability to AUC empirically in §G (Figures 10 & 12).

2. While a shared variance is a strong assumption, it is perhaps the simplest way of incorporating the inductive bias that $\Sigma_{g,g}$ will be highly correlated with the inverse of $n_g$, the number of samples from group $g$. In practice, we set $\sigma^2$ to be the pooled estimate $\frac{1}{n-d} \sum_{g=1}^{d} \sum_{z \in S \cap \mathcal{Z}_g} (f(z) - y_g)^2$.

3. Gaussian mean estimation is one of the best-studied problem in statistics, with numerous well-tested baselines and approaches for developing new methods. In particular, we make significant use of the classic James–Stein approach [James and Stein, 1961, Bock, 1975], SURE [Stein, 1981], and empirical Bayesian estimation methods [**?**].

4. In the multi-task setting, clients are likely to be unwilling to share their actual data but possibly more willing to share group summary statistics. Thus methods developed for our Gaussian model—which only require the group means $\mathbf{y}$, group counts $\mathbf{n}$, and an estimate of $\sigma^2$—will be more broadly applicable than methods that act directly on the dataset $S \subset \mathcal{Z}$.

---

[1] We assume existence of the first (and, in §2.2, of the second) moments of $f(z)$, $z \sim \mathcal{D} \mid \mathcal{Z}_g$, across all $g \in [d]$.
[2] If $n_g = 0$ we let $\hat{\mu}_g^{\text{naive}}$ fall back to pooling (Eq. 3), i.e., $\hat{\mu}_g^{\text{naive}} = \hat{\mu}_g^{\text{pooled}}$; in the next section, assume $n_g > 0 \; \forall \, g$.

This model can also be naturally extended—using a non-diagonal covariance $\mathbf{\Sigma}$—to disaggregated evaluation with non-disjoint groups, e.g., to simultaneously estimate performance both for all women and for only women in their forties. In the interest of brevity we focus on the disjoint group setting.

## 2.3 The multi-task setting

We can easily extend this model to study multi-task disaggregated evaluation, in which for each task $t = 1, \dots, T$ (e.g., a client of the model provider) we observe a set $S_t \subset \mathcal{Z}$ of $n_t$ samples from the task distribution $\mathcal{D}_t$. The goal is then to output $T$ vectors $\hat{\boldsymbol{\mu}}_t$ that are close on-average to the tasks' subpopulation errors $\mu_{t;g} = \mathbb{E}_{z \sim \mathcal{D}_t}[f(z)|z \in \mathcal{Z}_g]$. Converting to our Gaussian model, we observe $T$ vectors $\mathbf{y}_t \sim \mathcal{N}(\boldsymbol{\mu}_t, \mathbf{\Sigma}_t)$—where we set $\Sigma_{t;g,g} = \sigma^2/n_{t;g}$ for some globally shared $\sigma^2$ and task-specific group count vectors $\mathbf{n}_t \in \mathbb{Z}_{\geq 0}^d$—and must output $T$ mean estimates $\hat{\boldsymbol{\mu}}_t(\{\mathbf{y}_t\}_{t=1}^T) \in \mathbb{R}^d$.

We consider two natural multi-task baseline estimators. The first is the **global naive estimator** (or global estimator for short), which combines the data from all tasks, computes a single global vector of group averages, and uses it as the estimate for each task:

$$\hat{\boldsymbol{\mu}}_t^{\text{global}}(\{S_t\}_{t=1}^T) = \hat{\boldsymbol{\mu}}^{\text{naive}}\left(\bigcup_{t=1}^T S_t\right) \quad \text{or} \quad \hat{\boldsymbol{\mu}}_t^{\text{global}}(\{\mathbf{y}_t\}_{t=1}^T) = \left(\sum_{t=1}^T \mathbf{\Sigma}_t^{-1}\right)^{-1} \sum_{t=1}^T \mathbf{\Sigma}_t^{-1}\mathbf{y}_t. \quad (4)$$

While low variance, the global estimator ignores variation across tasks. Our second baseline—the **multi-task offset estimator**—shifts the global estimate on each task to ensure that the task's pooled mean is preserved (thus accounting for the variation in pooled means across individual tasks):

$$\hat{\boldsymbol{\mu}}_t^{\text{offset}}(\{\mathbf{y}_t\}_{t=1}^T) = \boldsymbol{\theta} + \hat{\boldsymbol{\mu}}^{\text{pooled}}(\mathbf{y}_t - \boldsymbol{\theta}), \qquad \text{where} \qquad \boldsymbol{\theta} = \hat{\boldsymbol{\mu}}_t^{\text{global}}(\{\mathbf{y}_t\}_{t=1}^T). \quad (5)$$

# 3 Methods

In the last section we reduced the problem of disaggregated evaluation to that of estimating a mean $\boldsymbol{\mu} \in \mathbb{R}^d$ given a sample $\mathbf{y} \sim \mathcal{N}(\boldsymbol{\mu}, \mathbf{\Sigma})$, where $\mathbf{\Sigma}$ is known and diagonal. We now design a method, **SureMap**, for the latter problem. Our technical approach involves the following two steps:

1. **Choosing a parameterized mean estimator.** We use the MAP estimator under a multivariate normal prior that we design specifically for intersectional subpopulations.
2. **Tuning the estimator's hyperparameters.** We use SURE to estimate the quality of our estimator, which we then optimize over the choice of hyperparameters using the L-BFGS-B algorithm.

## 3.1 Designing a parameterized estimator

As mean estimation is a vast area, we use three criteria for designing an estimator: it should (1) dominate baselines such as $\hat{\boldsymbol{\mu}}^{\text{naive}}$ and $\hat{\boldsymbol{\mu}}^{\text{pooled}}$; (2) have relatively few hyperparameters; and (3) handle *heteroskedasticity* stemming from variation in group sizes. One natural source of candidates are James–Stein-type shrinkage estimators: the original James–Stein estimator famously dominates $\hat{\boldsymbol{\mu}}^{\text{naive}}$ in MSE and has no hyperparameters to tune [James and Stein, 1961], satisfying our first two desiderata. Furthermore, while James and Stein [1961] assumed an isotropic $\mathbf{\Sigma}$, subsequent estimators such as the following variant of an estimator due to Bock [1975] do handle heteroskedastic $\mathbf{\Sigma}$:[3]

$$\hat{\boldsymbol{\mu}}_{\boldsymbol{\theta}}^{\text{Bock}}(\mathbf{y}) = \boldsymbol{\theta} + \left(1 - \frac{d-2}{(\mathbf{y}-\boldsymbol{\theta})^\top \mathbf{\Sigma}^{-1}(\mathbf{y}-\boldsymbol{\theta})}\right)_+ (\mathbf{y}-\boldsymbol{\theta}), \quad (6)$$

where $(\cdot)_+ = \max\{\cdot, 0\}$, and $\boldsymbol{\theta} \in \mathbb{R}^d$ is a default estimate towards which $\mathbf{y}$ is shrunk.[4]

However, empirically we find that this often underperforms the pooled estimator in low-sample regimes; further, the form of $\hat{\boldsymbol{\mu}}_{\boldsymbol{\theta}}^{\text{Bock}}$ shows that the amount of shrinkage towards $\boldsymbol{\theta}$ is the same for each coordinate $g \in [d]$, despite intuition suggesting that we should shrink less for groups $g$ with more samples. Corrections to this tend to be involved and difficult to generalize [Efron and Morris, 1973].

We thus turn to a different family of well-known Gaussian mean estimators: those that return the mode of the posterior distribution assuming $\boldsymbol{\mu}$ is sampled from the conjugate prior $\mathcal{N}(\boldsymbol{\theta}, \mathbf{\Lambda})$ with mean $\boldsymbol{\theta} \in \mathbb{R}^d$ and positive-definite covariance $\mathbf{\Lambda} \in \mathbb{R}^{d \times d}$ (e.g., Gelman et al. [2014, Equation 3.12]):

$$\hat{\boldsymbol{\mu}}_{\boldsymbol{\theta}, \mathbf{\Lambda}}^{\text{MAP}}(\mathbf{y}) = (\mathbf{\Lambda}^{-1} + \mathbf{\Sigma}^{-1})^{-1}(\mathbf{\Lambda}^{-1}\boldsymbol{\theta} + \mathbf{\Sigma}^{-1}\mathbf{y}). \quad (7)$$

---

[3]Feldman et al. [2014] show that using $d$ in the numerator instead of Bock's $\frac{\text{Tr}\,\mathbf{\Sigma}}{\|\mathbf{\Sigma}\|}$ performs better; in §B we show that their modified form can be derived by minimizing an upper bound on the $\mathbf{\Sigma}^{-1}$-weighted MSE.

[4]We typically use $\boldsymbol{\theta} = \mathbf{0}$, but in §B we derive a variant shrinking towards $\boldsymbol{\theta} = \hat{\boldsymbol{\mu}}^{\text{pooled}}(\mathbf{y})$.

**Algorithm 1:** Single-task SureMap.
(For multi-task SureMap see §D.)

---

**Input:** target $f : \mathcal{Z} \to \mathbb{R}$, samples $S \subset \mathcal{Z}$,
partition $\{\mathcal{Z}_g\}_{g=1}^d$ of $\mathcal{Z}$, each group $g$
an intersection of $k \in \mathbb{Z}_{>0}$ attributes

```
// compute naive group means
```
**for** *group* $g \in [d]$ **do**
  |   $n_g \leftarrow |S \cap \mathcal{Z}_g|$
  |   $y_g \leftarrow \frac{1}{n_g} \sum_{z \in S \cap \mathcal{Z}_g} f(z)$

```
// estimate group variances
```
$\sigma^2 \leftarrow \frac{1}{|S|-d} \sum_{g=1}^d \sum_{z \in S \cap Z_g} (f(z) - y_g)^2$
$\boldsymbol{\Sigma}^{-1} \leftarrow \mathrm{diag}(\mathbf{n})/\sigma^2$

```
// compute auxiliary matrix
```
**Method $\mathbf{A}(\boldsymbol{\tau})$:**
  |   `// compute prior covariance`
  |   (matrices $\mathbf{C}_A$ are defined in
  |   §3.1.1 and §C.1)
  |   $\boldsymbol{\Lambda} \leftarrow \sum_{A \subseteq [k]} \tau_A^2 \mathbf{C}_A$
  |   **Output:** $(\mathbf{I}_d + \boldsymbol{\Lambda}\boldsymbol{\Sigma}^{-1})^{-1}$

```
// optimize SURE using L-BFGS-B
```
$\hat{\boldsymbol{\tau}} \leftarrow \underset{\boldsymbol{\tau} \in \mathbb{R}_{\geq 0}^{2^k}}{\arg\min} \|\mathbf{A}(\boldsymbol{\tau})\mathbf{y}\|_{\boldsymbol{\Sigma}^{-1}}^2 - 2\,\mathrm{Tr}(\mathbf{A}(\boldsymbol{\tau}))$

```
// estimate group means using MAP
```
**Output:** $\mathbf{y} - \mathbf{A}(\hat{\boldsymbol{\tau}})\mathbf{y}$

---

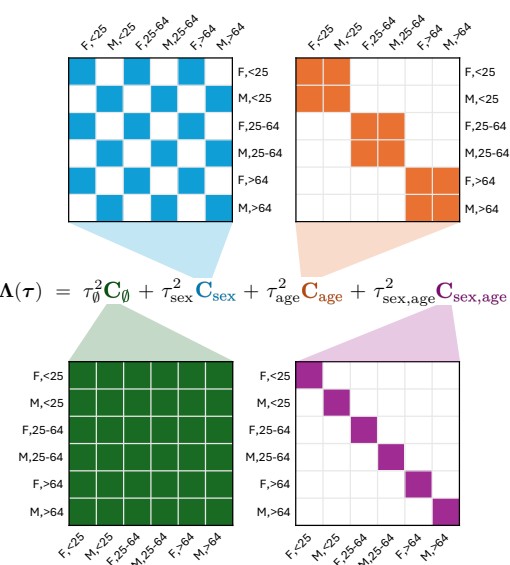

$$\boldsymbol{\Lambda}(\boldsymbol{\tau}) \;=\; \tau_\emptyset^2 \mathbf{C}_\emptyset + \tau_{\mathrm{sex}}^2 \mathbf{C}_{\mathrm{sex}} + \tau_{\mathrm{age}}^2 \mathbf{C}_{\mathrm{age}} + \tau_{\mathrm{sex,age}}^2 \mathbf{C}_{\mathrm{sex,age}}$$

Figure 1: Matrices $\mathbf{C}_A \in \{0,1\}^{d \times d}$, whose linear combination defines the covariance $\boldsymbol{\Lambda}(\boldsymbol{\tau})$ of our additive intersectional effects prior (Eq. 9). In this example, there are two categories for sex (`F`,`M`) and three for age (`<25`,`25-64`,`>64`), yielding $d = 6$ groups. Shaded squares are 1s, unshaded are 0s.

MAP naturally handles heteroskedasticity by weighting low-variance coordinates $g$ more heavily, satisfying our third criterion; however, its most general form has $\mathcal{O}(d^2)$ hyperparameters, violating the second. We next use problem structure to restrict the number of hyperparameters needed to define $\boldsymbol{\Lambda}$ while still allowing $\hat{\boldsymbol{\mu}}_{\boldsymbol{\theta},\boldsymbol{\Lambda}}^{\mathrm{MAP}}$ to express every baseline introduced in §2.1, satisfying our first criterion.

### 3.1.1 An additive intersectional effects prior

For brevity, we build up our estimator somewhat informally; a full description is in §C.1. We return to our simple example where each group $g \in [d]$ corresponds to an intersection $(s, a) \in [d_1] \times [d_2]$ of two attributes: a sex $s \in [d_1]$ and an age $a \in [d_2]$. A simple prior that additively incorporates individual attribute effects into intersectional group means is the following:

$$\mu_g = \tau_\emptyset \zeta + \tau_{\mathrm{sex}} \zeta_s^{\mathrm{sex}} + \tau_{\mathrm{age}} \zeta_a^{\mathrm{age}} + \tau_{\mathrm{sex,age}} \zeta_g + \theta_g \tag{8}$$

where $\boldsymbol{\theta} \in \mathbb{R}^d$ and $\tau_\emptyset, \tau_{\mathrm{sex}}, \tau_{\mathrm{age}}, \tau_{\mathrm{sex,age}} \in \mathbb{R}_{\geq 0}$ are hyperparameters, $\zeta \sim \mathcal{N}(0,1)$ is a scalar effect shared across all groups, the vector $\boldsymbol{\zeta}^{\mathrm{sex}} \in \mathbb{R}^{d_1}$ has its $s$th entry $\zeta_s^{\mathrm{sex}} \sim \mathcal{N}(0,1)$ shared by all groups $g$ whose sex is $s$, the vector $\boldsymbol{\zeta}^{\mathrm{age}} \in \mathbb{R}^{d_2}$ has its $a$th entry $\zeta_a^{\mathrm{age}} \sim \mathcal{N}(0,1)$ shared by all groups $g$ whose age is $a$, and the vector $\boldsymbol{\zeta} \sim \mathcal{N}(\mathbf{0}_d, \mathbf{I}_d)$ contains an independent noise term $\zeta_g$ for each group $g$.

The hyperparameter $\tau_\emptyset$ quantifies how much we expect all of the means to be shifted (by a shared positive or negative value) from the default $\boldsymbol{\theta}$. Hyperparameters $\tau_{\mathrm{sex}}$ and $\tau_{\mathrm{age}}$ express how large we expect contributions of sex and age alone to be towards the means. And finally, non-zero $\tau_{\mathrm{sex,age}}$ gives the prior flexibility to model heterogeneity across all intersectional groups.

Given the vector of hyperparameters $\boldsymbol{\tau} = (\tau_\emptyset, \ldots, \tau_{\mathrm{sex,age}}) \in \mathbb{R}_{\geq 0}^4$, the prior can be written more compactly as $\boldsymbol{\mu} \sim \mathcal{N}(\boldsymbol{\theta}, \boldsymbol{\Lambda}(\boldsymbol{\tau}))$, where the covariance is

$$\boldsymbol{\Lambda}(\boldsymbol{\tau}) = \sum_{A \subseteq \{\mathrm{sex,age}\}} \tau_A^2 \mathbf{C}_A = \tau_\emptyset^2 \mathbf{C}_\emptyset + \tau_{\mathrm{sex}}^2 \mathbf{C}_{\mathrm{sex}} + \tau_{\mathrm{age}}^2 \mathbf{C}_{\mathrm{age}} + \tau_{\mathrm{sex,age}}^2 \mathbf{C}_{\mathrm{sex,age}} \tag{9}$$

for matrices $\mathbf{C}_A \in \{0,1\}^{d \times d}$ s.t. each entry $C_{A;g,h}$ is one iff groups $g$ and $h$ agree on the attributes included in $A$. In particular, we have that $\mathbf{C}_\emptyset = \mathbf{1}_{d \times d}$ is the all-ones matrix, the entries $C_{\mathrm{sex};g,h}$ of $\mathbf{C}_{\mathrm{sex}} \in \{0,1\}^{d \times d}$ are one iff groups $g$ and $h$ share the same sex attribute, the matrix $\mathbf{C}_{\mathrm{age}}$ is analogous, and $\mathbf{C}_{\mathrm{sex,age}} = \mathbf{I}_d$ is the $d \times d$ identity. This structure is visualized in Figure 1.

### 3.1.2 Efficiency and expressivity

As detailed in §C.1, this prior can be naturally extended to any number of attributes $k$ using a covariance matrix $\mathbf{\Lambda}(\boldsymbol{\tau}) \in \mathbb{R}^{d \times d}$ specified by a vector $\boldsymbol{\tau} \in \mathbb{R}_{\geq 0}^{2^k}$ of $2^k$ hyperparameters. Since $k \leq \lfloor \log_2 d \rfloor$, the total number of hyperparameters (including $\boldsymbol{\theta} \in \mathbb{R}^d$) is $d + 2^k = \mathcal{O}(d)$, which is much smaller than the $\mathcal{O}(d^2)$ complexity of the general case. We can further reduce this by fixing the entries of $\boldsymbol{\theta}$, constraining them to be identical, or setting them using external (e.g., multi-task) data.

Despite this reduction in hyperparameters, we can show that for a suitable choice of $\boldsymbol{\tau}$, the estimator $\hat{\boldsymbol{\mu}}_{\boldsymbol{\theta}, \mathbf{\Lambda}(\boldsymbol{\tau})}^{\text{MAP}}$ recovers many estimators of interest, including the naive estimator and the (possibly offset) pooled estimator (see §C.2). This means that MAP with our structured prior should be able to outperform all four baselines from the previous section, if appropriately tuned.

## 3.2 Tuning by minimizing expected risk

Having specified a parameterized estimator, there remains the question of setting its parameters $\boldsymbol{\theta}$ and $\boldsymbol{\tau}$. One might want to treat this as a hyperparameter tuning problem and use a data-splitting approach; however, the dimensionality of the problem makes standard techniques either expensive or noisy, and data splitting introduces additional randomness and design decisions into an already data-poor environment. We instead make continued use of our Gaussian assumption and turn to SURE, which given a differentiable estimator $\hat{\boldsymbol{\mu}} : \mathbb{R}^d \to \mathbb{R}^d$ returns an unbiased estimate of its weighted MSE $L_{\boldsymbol{\mu}}^{\mathbf{n}}$ using sample data $\mathbf{y} \sim \mathcal{N}(\boldsymbol{\mu}, \boldsymbol{\Sigma})$:

$$\hat{R}_{\boldsymbol{\mu}}^{\mathbf{n}}(\mathbf{y}) = \frac{\sigma^2}{d} \left( \|\hat{\boldsymbol{\mu}}(\mathbf{y}) - \mathbf{y}\|_{\boldsymbol{\Sigma}^{-1}}^2 - d + 2 \boldsymbol{\nabla}_{\mathbf{y}} \cdot \hat{\boldsymbol{\mu}}(\mathbf{y}) \right), \tag{10}$$

where given any $\mathbf{W} \succ \mathbf{0}_{d \times d}$ we denote $\|\mathbf{x}\|_{\mathbf{W}}^2 = \langle \mathbf{x}, \mathbf{W}\mathbf{x} \rangle \; \forall \; \mathbf{x} \in \mathbb{R}^d$. Using SURE we can now tune the parameters of $\hat{\boldsymbol{\mu}}$ by minimizing $\hat{R}_{\boldsymbol{\mu}}^{\mathbf{n}}(\mathbf{y})$ in a manner similar to empirical risk minimization.

### 3.2.1 Single-task SureMap

In the single-task setting, we fix $\boldsymbol{\theta} = \mathbf{0}_d$ and tune the variance parameters $\boldsymbol{\tau} \in \mathbb{R}_{\geq 0}^{2^k}$. Letting $\mathbf{A}(\boldsymbol{\tau}) = (\mathbf{\Lambda}^{-1}(\boldsymbol{\tau}) + \boldsymbol{\Sigma}^{-1})^{-1} \mathbf{\Lambda}^{-1}(\boldsymbol{\tau})$, we define the **single-task SureMap estimator** as

$$\hat{\boldsymbol{\mu}}^{\text{SM}}(\mathbf{y}) = \hat{\boldsymbol{\mu}}_{\mathbf{0}_d, \mathbf{\Lambda}(\hat{\boldsymbol{\tau}})}^{\text{MAP}}(\mathbf{y}) = (\mathbf{I}_d - \mathbf{A}(\hat{\boldsymbol{\tau}}))\mathbf{y} \tag{11}$$

$$\text{for } \hat{\boldsymbol{\tau}} = \arg\min_{\boldsymbol{\tau} \in \mathbb{R}_{\geq 0}^{2^k}} \|\mathbf{A}(\boldsymbol{\tau})\mathbf{y}\|_{\boldsymbol{\Sigma}^{-1}}^2 - 2 \operatorname{Tr}(\mathbf{A}(\boldsymbol{\tau})). \tag{12}$$

The optimization problem in the second line comes from substituting $\hat{\boldsymbol{\mu}}_{\mathbf{0}_d, \mathbf{\Lambda}(\hat{\boldsymbol{\tau}})}^{\text{MAP}}$ into SURE (Eq. 10). It is nonconvex, but we find that it can be quickly solved to sufficient accuracy with L-BFGS-B [Byrd et al., 1995], a standard method for bound-constrained optimization of differentiable functions.

### 3.2.2 Multi-task SureMap

To generalize SureMap to the multi-task setting we propose to specify $\hat{\boldsymbol{\theta}}$ and $\hat{\boldsymbol{\tau}}$ by minimizing SURE aggregated across tasks, i.e., $\sum_{t=1}^{T} \hat{R}_{\boldsymbol{\mu}_t}^{\mathbf{n}_t}(\mathbf{y}_t)$. While setting both parameters via direct optimization of this objective is the most straightforward approach, we find that it performs worse than single-task SureMap when there are only a few tasks ($T \leq 5$) and rarely improves significantly above the multi-task global and offset estimators. This can be explained by observing the few-task limit—i.e. $T = 1$—in which case optimizing the aggregated SURE objective results in setting $\hat{\boldsymbol{\theta}} = \mathbf{y}_1$ and thus makes the multi-task estimator equivalent to the naive estimator.

We find that a better approach is to treat the choice of $\hat{\boldsymbol{\theta}}$ as its own simultaneous mean estimation problem and apply the SureMap approach to it. In particular, our model $\mathbf{y}_t \sim \mathcal{N}(\boldsymbol{\mu}_t, \boldsymbol{\Sigma}_t)$ and our prior $\boldsymbol{\mu}_t \sim \mathcal{N}(\boldsymbol{\theta}, \mathbf{\Lambda})$ imply that the samples $\mathbf{y}_t \sim \mathcal{N}(\boldsymbol{\theta}, \mathbf{\Lambda} + \boldsymbol{\Sigma}_t)$ have mean $\boldsymbol{\theta}$ and known covariances (apart from tuning parameters). Therefore, the MAP estimator of $\boldsymbol{\theta}$ itself given a hyperprior $\boldsymbol{\theta} \sim \mathcal{N}(\mathbf{0}_d, \boldsymbol{\Gamma})$ with covariance $\boldsymbol{\Gamma} \succ \mathbf{0}$ will have the form $\hat{\boldsymbol{\theta}} = \left(\boldsymbol{\Gamma}^{-1} + \sum_{t=1}^{T}(\mathbf{\Lambda} + \boldsymbol{\Sigma}_t)^{-1}\right)^{-1} \sum_{t=1}^{T}(\mathbf{\Lambda} + \boldsymbol{\Sigma}_t)^{-1}\mathbf{y}_t$.

To reduce the number of tuning parameters, we use a prior of the same form as before by specifying $\boldsymbol{\Gamma} = \mathbf{\Lambda}(\boldsymbol{\upsilon})$ for $\boldsymbol{\upsilon} \in \mathbb{R}_{\geq 0}^{2^k}$, i.e., the same structured covariance as described in §3.1.1 but with separately tuned parameters (see §3.1.1 and §C.1). Substituting the meta-level MAP estimator of $\boldsymbol{\theta}$ into the MAP estimator of $\boldsymbol{\mu}_t$ and tuning the parameters $\boldsymbol{\tau}$ and $\boldsymbol{\upsilon}$ by optimizing the sum of SUREs (Eq. 10)

across tasks yields our **multi-task SureMap estimator** (see §C.3 for details):

$$\hat{\boldsymbol{\mu}}_t^{\mathrm{SM}}(\{\mathbf{y}_t\}_{t=1}^T) = \hat{\boldsymbol{\mu}}_{\hat{\boldsymbol{\theta}}(\hat{\boldsymbol{\tau}},\hat{\boldsymbol{v}}),\boldsymbol{\Lambda}(\hat{\boldsymbol{\tau}})}^{\mathrm{MAP}}(\mathbf{y}_t) = \mathbf{y}_t + \mathbf{A}_t(\hat{\boldsymbol{\tau}})(\hat{\boldsymbol{\theta}}(\hat{\boldsymbol{\tau}},\hat{\boldsymbol{v}}) - \mathbf{y}_t) \tag{13}$$

$$\text{for } \hat{\boldsymbol{\theta}}(\boldsymbol{\tau},\boldsymbol{v}) = \sum_{t=1}^T \mathbf{M}_t(\boldsymbol{\tau},\boldsymbol{v})\mathbf{y}_t, \qquad \mathbf{A}_t(\boldsymbol{\tau}) = \left(\boldsymbol{\Lambda}^{-1}(\boldsymbol{\tau}) + \boldsymbol{\Sigma}_t^{-1}\right)^{-1}\boldsymbol{\Lambda}^{-1}(\boldsymbol{\tau}),$$

$$\mathbf{M}_t(\boldsymbol{\tau},\boldsymbol{v}) = \left(\boldsymbol{\Lambda}^{-1}(\boldsymbol{v}) + \sum_{s=1}^T (\boldsymbol{\Lambda}(\boldsymbol{\tau}) + \boldsymbol{\Sigma}_s)^{-1}\right)^{-1}(\boldsymbol{\Lambda}(\boldsymbol{\tau}) + \boldsymbol{\Sigma}_t)^{-1},$$

$$\hat{\boldsymbol{\tau}},\hat{\boldsymbol{v}} = \underset{\boldsymbol{\tau},\boldsymbol{v} \in \mathbb{R}_{\geq 0}^{2k}}{\arg\min} \sum_{t=1}^T \left[\left\|\mathbf{A}_t(\boldsymbol{\tau})(\hat{\boldsymbol{\theta}}(\boldsymbol{\tau},\boldsymbol{v}) - \mathbf{y}_t)\right\|_{\boldsymbol{\Sigma}_t^{-1}}^2 + 2\operatorname{Tr}\left(\mathbf{A}_t(\boldsymbol{\tau})(\mathbf{M}_t(\boldsymbol{\tau},\boldsymbol{v}) - \mathbf{I}_d)\right)\right]. \tag{14}$$

The optimization problem on the last line can again be approximately solved using L-BFGS-B.

## 3.3 Limitations

Various modeling assumptions impact performance of SureMap. For instance, the Gaussian error assumption is less appropriate when errors are heavy-tailed. In §G, Figure 9, we consider MSE as an example of a target metric with heavy-tailed observation errors. We find that SureMap still performs well, but is no longer superior to previous approaches. One avenue for improvement would be to use a variance-stabilizing transformation (e.g., Hawkins and Wixley [1986]) prior to applying SureMap.

Note that SureMap achieves its improved accuracy by shrinking naive estimates towards a less granular estimator (e.g., a pooled mean). As a result the estimation is biased towards *less disparity*, which could lead to overly optimistic conclusions about fairness. For this reason it is extremely important to examine not just the point estimates, but also confidence intervals. These can be obtained, for example, by viewing SureMap as a regression approach (§E) and leveraging inference techniques for regression.

## 4 Datasets

We evaluate our approach in several representative settings for disaggregated evaluation, including two tabular settings appearing in previous works [Miller et al., 2021, Herlihy et al., 2024, Liu et al., 2024], and three new settings: a multi-task tabular setting based on state-level U.S. census data and both a single-task and a multi-task ASR evaluation setting.

### 4.1 Tabular datasets

We consider three tabular datasets, two for the single-task and one for the multi-task setting, covering two important domains where fairness concerns can arise: healthcare records and demographic data. While we focus on the classification task and 0-1 error, in §G we also report results for regression (Figures 8 & 9) and for classification with AUC as the target metric (Figures 10 & 12).

**Diabetes.** This is a tabular dataset of Strack et al. [2014], containing around 100K patient records with six race, two sex, and three age categories. We evaluate a logistic regression classifier trained to predict patient disposition after a hospital stay (discharged or otherwise). The target metric is the 0-1 error.

**Adult.** We use the classic *Adult* census dataset [Kohavi, 1996] to evaluate performance of an in-context LLM learner—specifically `llama-3-70b`—in predicting whether a person makes more or less than $50K after being provided with eight examples via a modification of the prompt template of Liu et al. [2024]. The target metric is the 0-1 error, disaggregation is by race, sex, and age.

**State-Level ACS (SLACS).** This is a tabular dataset for multi-task setting derived from the census data for all U.S. states and Puerto Rico assembled by Ding et al. [2021]. Each datapoint corresponds to a person in one of nine race and two sex categories; we consider three age categories: below 25, 25–64, and over 64. The underlying task is to classify each person as earning either more or less than $50K. We train a regularized logistic model on the data from California, and seek to evaluate its performance on the other 50 states/territories, which comprise the tasks. The target metric is the 0-1 error.

### 4.2 ASR datasets

We also introduce both single-task and multi-task *speech recognition* datasets, based on applying the popular *Whisper* ASR model [Radford et al., 2023]—specifically `whisper-tiny`—on the English part of the *Common Voice* (CV) dataset [Ardila et al., 2020], which contains utterances from individuals in one of nine age and three sex categories.

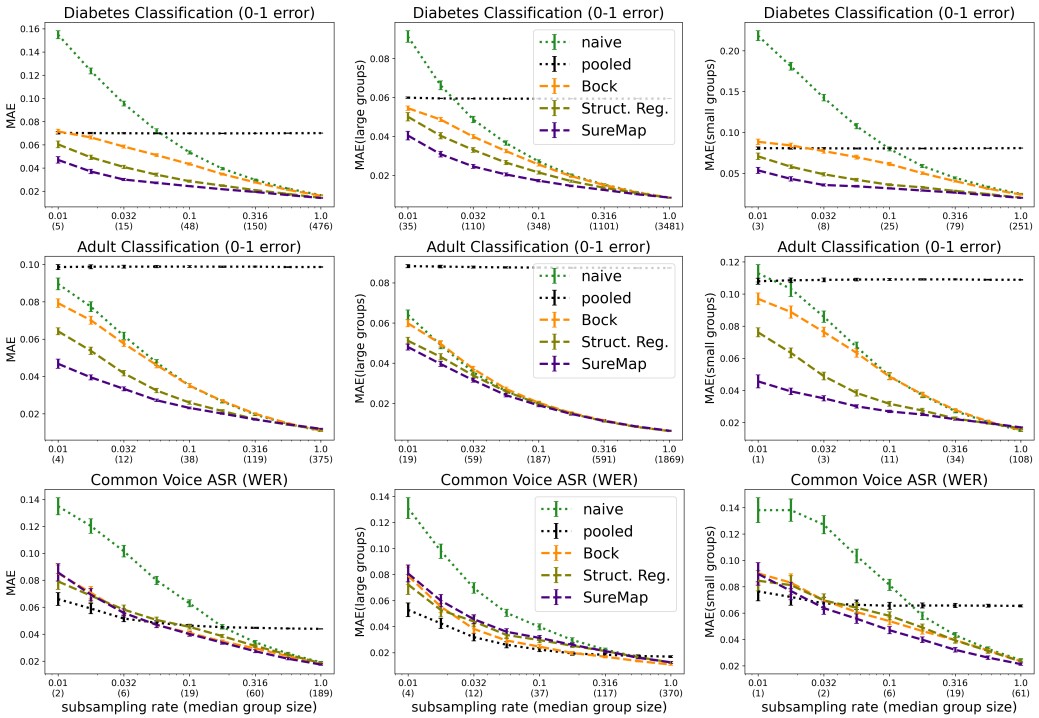

Figure 2: Single-task evaluations on Diabetes (top, disaggregating by race, sex, and age), Adult (middle, disaggregating by race, sex, and age), and Common Voice (bottom, disaggregating by sex and age). The MAE is averaged across all groups (left), large groups (center), or small groups (right). Large and small groups are defined as above and below median group size.

**Common Voice.** This is a single-task dataset obtained by combining the validation and test partitions of the CV dataset. We calculate the word-error rate (WER) across all the utterances of each individual, which becomes the target metric to be predicted.

**Common Voice Clusters (CVC).** This is a multi-task ASR dataset. To construct it, we first cluster the utterances in the train partition of the CV dataset into 20 clusters by applying $k$-means to the sums of GloVe word embeddings [Pennington et al., 2014] of their corresponding text strings. To model task relatedness, we then randomly reassign each utterance to a random cluster with probability $\alpha \in [0, 1]$. The resulting clusters are the tasks. The target metric is the word-error rate (WER) across all the utterances of each individual in a given cluster. In most experiments we use $\alpha = \frac{1}{2}$, but we also investigate what happens when $\alpha$ varies between zero, i.e., the original clusters, and one, corresponding to identically distributed tasks.

## 5 Evaluation

Our main metric is MAE relative to a ground truth vector, which we take to be the mean of all available data for each subpopulation $g \in [d]$, except those with fewer than 40 samples. In our main results we subsample with replacement from the entire dataset at different rates and track performance as a function of the sizes of the resulting datasets. To obtain 95% confidence intervals we conduct 200 and 40 random trials at each subsampling rate in the single-task and multi-task settings, respectively.

### 5.1 Single-task

We compare SureMap to the naive (Eq. 2) and pooled (Eq. 3) baselines, as well as to the Bock estimator with shrinkage towards the pooled estimator (Eq. 17) and the structured regression estimator of Herlihy et al. [2024]. On both Diabetes and Adult, SureMap significantly outperforms all competitors (Figure 2, top & middle), the greatest improvement is on subpopulations with limited data. In §G, we consider a regression variant of Diabetes and observe similar results (Figure 8). On the Common Voice task, SureMap performs roughly similarly to Bock, while outperforming structured regression at some subsampling rates (Figure 2, bottom); here again the gains are driven

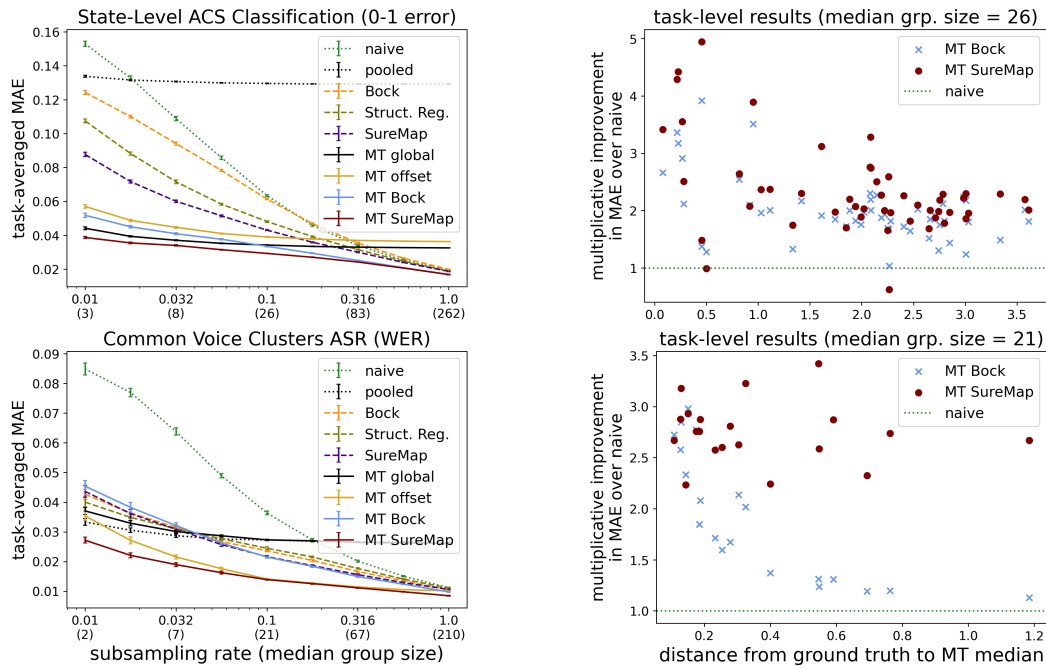

Figure 3: Multi-task evaluations on SLACS (top, disaggregating by race, sex, and age) and CVC (bottom, disaggregating by sex and age). *Left:* Performance across different subsampling rates. *Right:* Multiplicative improvement in MAE over naive estimator on individual tasks; subsampling rate=0.1.

by better performance on small groups. Pooling performs best when data is extremely limited, but it is not competitive with even modestly more data, and also underperforms on small groups.

## 5.2 Multi-task

In the multi-task setting, we use all the single-task methods as baselines while adding multi-task (MT) ones, including MT global (Eq. 4), MT offset (Eq. 5), and an MT extension of Bock (Eq. 6) in which $\theta_g$ is set using the average across the group $g$ data on all *other* tasks. We first consider the SLACS task, for which Figure 3 (top left) shows that MT SureMap significantly outperforms other methods in the low-data regime while matching the best one (MT Bock) in the high-data regime. At subsampling rate 0.1, Figure 3 (top right) also shows that using multi-task data leads to improvement on all but two of the fifty tasks, that the reduction in MAE over the naive estimator on a typical task is 2x, and that this improvement only loosely correlates with the task's ground truth distance from the multi-task median. On the other hand, it also shows that while MT Bock's improvements are typically smaller, on SLACS it improves performance for *every* state (including the two where MT SureMap is worse).

On the CVC task, the MT offset baseline is the most competitive, except at the lowest subsampling rate where pooling is better and at higher subsampling rates where it stops improving with additional data (Figure 3, bottom left). SureMap outperforms it and all other methods across all subsampling rates and its advantage is greatest in low-data regimes, where it even outperforms pooling. In the task-level evaluation in Figure 3 (bottom right) we see that on every task, MT SureMap attains an improvement of 2–3.5x over the naive baseline *and* almost always outperforms MT Bock. Furthermore, the latter performs substantially worse on tasks whose ground truth vectors are far away from the multi-task center while MT SureMap is not affected.

## 5.3 Ablations

We next look at how the degree of included intersectional effects and multi-task structure affect performance. In Figure 4 (left), we evaluate utility of including higher-order interactions in the structured prior. We implement SureMap variants with up to $\ell$th-order interactions by setting $\tau_A$ for $|A| \in \{\ell+1, \ldots, k-1\}$ to zero (but not for $|A| = k$). We observe that including zeroth-order effects ($\ell = 0$) in single-task SureMap (i.e., shrinkage to pooling) improves upon the usual single-parameter Gaussian prior ($\ell = -1$) in low-data regimes. Adding first-order effects ($\ell = 1$) leads to substantial further im-

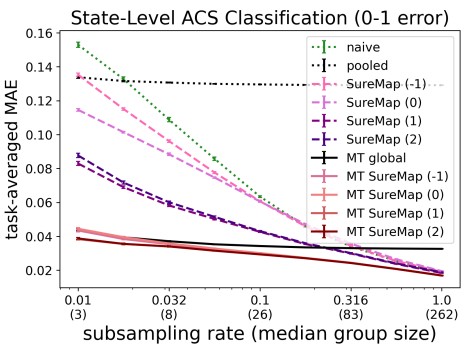
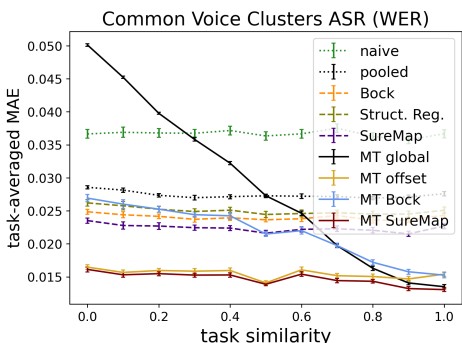

Figure 4: *Left:* Comparison of SureMap variants on SLACS. The SureMap ($\ell$) variant sets to zero the entries of $\boldsymbol{\tau}$ corresponding to interactions of size $> \ell$ (except for the highest-order interactions). *Right:* Evaluation of different methods as the interpolation coefficient that defines the CVC tasks is varied.

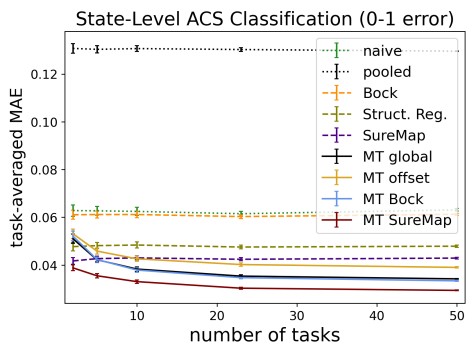
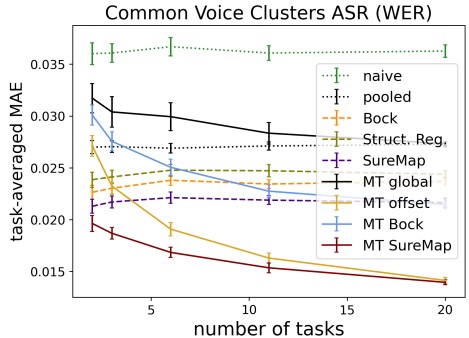

Figure 5: Performance as the number of tasks varies, evaluated on SLACS (left) and CVC (right).

provement, but second-order effects ($\ell = 2$) make performance slightly worse. A similar effect can be observed among the multi-task variants, except there is no loss (but also no gain) to using the highest-order variant ($\ell = 2$). Overall, this study suggests that using the full-order SureMap is a reasonable default but that most of the method's effectiveness comes from zeroth- and first-order effects.

Figure 4 (right) tracks performance as the task-similarity parameter defining the CVC task is varied. MT SureMap outperforms all methods at all settings and is also not as strongly affected by the task similarity, at least as it is defined for the CVC data. This suggests that the structured prior we use may be useful even if the dataset means are quite different but the underlying evaluation problem (in this case estimating WER of ASR models) is the same.

Lastly, in Figure 5 we study how the number of tasks affects multi-task performance. On both SLACS and CVC, MT SureMap outperforms all single-task baselines (the best one being the single-task SureMap) at $T = 2$ tasks, i.e., it can take advantage of even very little external information. In contrast, on CVC, the competitor multi-task methods (e.g., MT offset and MT Bock) do not even outperform single-task methods until $T \geq 5$ tasks. These results demonstrate that, unlike these comparators, multi-task SureMap can be confidently used even when only one additional client's worth of data is available.

## 6 Conclusion

We have introduced SureMap, a disaggregated evaluation approach, which combines MAP estimation under a structured Gaussian prior with hyperparameter tuning via SURE. SureMap achieves substantial empirical improvements over strong baselines in both single-task and multi-task settings. Valuable future directions include improving robustness to heavy-tailed data and developing multi-task methods that can handle client privacy concerns. More broadly, we hope our work will have a positive impact by allowing model users to more accurately identify fairness-related harms, the first step towards mitigating them. However, using SureMap and any other disaggregated evaluation approach must be done with care, so as to not risk overconfidence in a model's fairness.

## Acknowledgments

We thank Nicolas Le Roux for discussions and feedback at the beginning of this effort.

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

# A  Notation

- For $p \in [1, \infty]$ we use $\|\cdot\|_p$ to denote the $p$-norm on $\mathbb{R}^d$.
- We use $\|\cdot\|$ to denote the spectral norm on $\mathbb{R}^{m \times n}$.
- For any positive semi-definite matrix $\mathbf{A} \succeq \mathbf{0}_{d \times d}$ we use the notation $\|\cdot\|_{\mathbf{A}} : \mathbb{R}^d \to \mathbb{R}_{\geq 0}$ to denote the vector norm $\|\mathbf{x}\|_{\mathbf{A}} = \sqrt{\langle \mathbf{x}, \mathbf{A}\mathbf{x} \rangle} = \|\sqrt{\mathbf{A}}\mathbf{x}\|_2$ on $\mathbf{x} \in \mathbb{R}^d$.
- For any positive integer $k$ we use $[k]$ to denote the set $\{1, \ldots, k\}$.
- For any set $S$ we use $2^S$ to denote its powerset.
- For any vector $\mathbf{a} \in \mathbb{R}^d$ we use $a_i$ to denote its $i$th entry. If $\mathbf{a}$ is only defined as an expression then we will abuse notation and use $(\mathbf{a})_i$ to refer to its $i$th entry.
- For any vector $\mathbf{a} \in \mathbb{R}^k$ and any subset $S \subset [k]$ with elements $s_1 < \cdots < s_m$ we define $\mathbf{a}_S = \begin{pmatrix} a_{s_1} & \cdots & a_{s_m} \end{pmatrix}$ to be the vector of the entries of $\mathbf{a}$ whose indices correspond to the elements of $S$ sorted in ascending order.
- For any subset $S \subset [k]$ we assume $\bigtimes_{s \in S}$ iterates over the elements in ascending order.
- For any matrix $\mathbf{A} \in \mathbb{R}^{m \times n}$ we use $A_{i,j}$ to denote its $(i,j)$th entry and $\mathbf{A}_{i,:}$ its $i$th row. If $\mathbf{A}$ is only defined as an expression then we will abuse notation and use $(\mathbf{A})_{i,j}$ to refer to its $(i,j)$th entry.
- For any $k$-tensor $\mathbf{Z} \in \mathbb{R}^{\times_{a=1}^k d_a}$ with dimensions $d_1, \ldots, d_k \in \mathbb{Z}_{>0}$ and any vector $\mathbf{c} \in \bigtimes_{a=1}^k [d_a]$ of indices we use $Z_{\mathbf{c}}$ to refer the the $(c_1, \ldots, c_k)$th entry of $\mathbf{Z}$.
- We use $\mathbf{0}_m, \mathbf{1}_m \in \mathbb{R}^m$ and $\mathbf{0}_{m \times n}, \mathbf{1}_{m \times n} \in \mathbb{R}^{m \times n}$ to refer to all-zero and all-one vectors and matrices, and $\mathbf{I}_n$ to refer the $n \times n$ identity matrix.

# B  Heteroskedastic James–Stein-type shrinkage estimation

For reference we state a weighted version of Stein's unbiased risk estimate (SURE) [Stein, 1981]:

**Lemma B.1.** *Suppose* $\mathbf{y} \sim \mathcal{N}(\boldsymbol{\mu}, \boldsymbol{\Sigma})$ *for mean* $\boldsymbol{\mu} \in \mathbb{R}^d$ *and diagonal p.s.d. covariance* $\boldsymbol{\Sigma} \in \mathbb{R}^{d \times d}$, *and consider a function* $\hat{\boldsymbol{\mu}} : \mathbb{R}^d \to \mathbb{R}^d$ *s.t. for every* $i \in [d]$ *the function* $\hat{\mu}_i$ *is almost differentiable in* $y_i$ *and we have* $\mathbb{E}\|\boldsymbol{\nabla}_{\mathbf{y}} \cdot \hat{\boldsymbol{\mu}}(\mathbf{y})\|_1 < \infty$. *Then for any diagonal matrix* $\mathbf{W} \in \mathbb{R}^{d \times d}$ *we have* $\mathbb{E}\|\hat{\boldsymbol{\mu}}(\mathbf{y}) - \boldsymbol{\mu}\|_{\mathbf{W}}^2 = \mathbb{E}\|\hat{\boldsymbol{\mu}}(\mathbf{y}) - \mathbf{y}\|_{\mathbf{W}}^2 - \mathrm{Tr}(\mathbf{W}\boldsymbol{\Sigma}) + 2\mathbb{E}[\boldsymbol{\nabla}_{\mathbf{y}} \cdot (\mathbf{W}\boldsymbol{\Sigma}\hat{\boldsymbol{\mu}}(\mathbf{y}))]$.

*Proof.* Expanding the squared norm and applying Stein's Lemma [Stein, 1981, Lemma 2] to the last term yields

$$\begin{aligned} \mathbb{E}\|\hat{\boldsymbol{\mu}}(\mathbf{y}) - \boldsymbol{\mu}\|_{\mathbf{W}}^2 &= \mathbb{E}\|\hat{\boldsymbol{\mu}}(\mathbf{y}) - \mathbf{y}\|_{\mathbf{W}}^2 + \mathbb{E}\|\mathbf{y} - \boldsymbol{\mu}\|_{\mathbf{W}}^2 + 2\mathbb{E}\langle \hat{\boldsymbol{\mu}}(\mathbf{y}) - \mathbf{y}, \mathbf{W}(\mathbf{y} - \boldsymbol{\mu}) \rangle \\ &= \mathbb{E}\|\hat{\boldsymbol{\mu}}(\mathbf{y}) - \mathbf{y}\|_{\mathbf{W}}^2 + \mathrm{Tr}(\mathbf{W}\boldsymbol{\Sigma}) + 2\mathbb{E}[\boldsymbol{\nabla}_{\mathbf{y}} \cdot (\mathbf{W}\boldsymbol{\Sigma}\hat{\boldsymbol{\mu}}(\mathbf{y})) - 2\,\mathrm{Tr}(\mathbf{W}\boldsymbol{\Sigma})] \end{aligned} \tag{15}$$

$\square$

We now turn to James–Stein-type estimators of the form $\hat{\boldsymbol{\mu}}(\mathbf{y}) = \mathbf{y} - \frac{c}{\|\mathbf{P}(\mathbf{y}-\mathbf{b})\|_{\boldsymbol{\Sigma}^{-1}}^2}\mathbf{P}(\mathbf{y} - \mathbf{b})$ for $c \in \mathbb{R}$, $\boldsymbol{\theta} \in \mathbb{R}^d$, and $\mathbf{P} \in \mathbb{R}^{d \times d}$, roughly extending the one considered by Bock [1975] (in a more general form) to $\mathbf{P} \neq \mathbf{I}_d$ and $\mathbf{b} \neq \mathbf{0}_d$. Setting $\mathbf{x} = \mathbf{P}(\mathbf{y} - \boldsymbol{\theta})$, we apply Lemma B.1 to show that the expected $\boldsymbol{\Sigma}^{-1}$-weighted MSE is

$$\begin{aligned} \mathbb{E}\|\hat{\boldsymbol{\mu}}(\mathbf{y}) - \boldsymbol{\mu}\|_{\boldsymbol{\Sigma}^{-1}}^2 &= d + \mathbb{E}\left[ \frac{c^2 - 2c\,\mathrm{Tr}(\mathbf{P})}{\|\mathbf{x}\|_{\boldsymbol{\Sigma}^{-1}}^2} + \frac{4c\,\mathrm{Tr}(\mathbf{x}\mathbf{x}^\top \boldsymbol{\Sigma}^{-1}\mathbf{P})}{\|\mathbf{x}\|_{\boldsymbol{\Sigma}^{-1}}^4} \right] \\ &= d + \mathbb{E}\left[ \frac{c^2 - 2c\,\mathrm{Tr}(\mathbf{P})}{\|\mathbf{x}\|_{\boldsymbol{\Sigma}^{-1}}^2} + \frac{4c\,\mathrm{Tr}(\boldsymbol{\Sigma}^{-\frac{1}{2}}\mathbf{x}\mathbf{x}^\top \boldsymbol{\Sigma}^{-\frac{1}{2}}\boldsymbol{\Sigma}^{-\frac{1}{2}}\mathbf{P}\boldsymbol{\Sigma}^{\frac{1}{2}})}{\|\mathbf{x}\|_{\boldsymbol{\Sigma}^{-1}}^4} \right] \\ &\leq d + \mathbb{E}\left[ \frac{c^2 \|\boldsymbol{\Sigma}\| - 2c\,\mathrm{Tr}(\boldsymbol{\Sigma}\mathbf{P}) + 4c\|\boldsymbol{\Sigma}^{-\frac{1}{2}}\mathbf{P}\boldsymbol{\Sigma}^{\frac{1}{2}}\|}{\|\mathbf{x}\|_{\boldsymbol{\Sigma}^{-1}}^2} \right] \end{aligned} \tag{16}$$

where to compute the derivative in SURE we made use of the matrix calculus tool of Laue et al. [2018] and the last line follows by von Neumann's trace inequality. This upper bound is minimized at $c = \mathrm{Tr}(\mathbf{P}) - 2\|\boldsymbol{\Sigma}^{-\frac{1}{2}}\mathbf{P}\boldsymbol{\Sigma}^{\frac{1}{2}}\|$. Note that for $\mathbf{P} = \mathbf{I}_d$ this setting of $c$ exactly recovers the Bock estimator $\hat{\boldsymbol{\mu}}_{\theta}^{\mathrm{Bock}}$ in Equation 6, apart from the positive-part correction.

On the other hand, for shrinking to the pooled mean $\hat{\boldsymbol{\mu}}^{\text{pooled}}(\mathbf{y}) = \frac{\mathbf{1}_{d \times d} \boldsymbol{\Sigma}^{-1} \mathbf{y}}{\text{Tr}(\boldsymbol{\Sigma}^{-1})}$ we can apply a well-known fact about the eigenvalues of symmetric rank-one updates (e.g. Bunch et al. [1978, Theorem 1]) to diagonal matrices to obtain $\|\boldsymbol{\Sigma}^{-\frac{1}{2}} \mathbf{P} \boldsymbol{\Sigma}^{\frac{1}{2}}\| = \left\| \mathbf{I}_d - \frac{\boldsymbol{\Sigma}^{-\frac{1}{2}} \mathbf{1}_{d \times d} \boldsymbol{\Sigma}^{-\frac{1}{2}}}{\text{Tr}(\boldsymbol{\Sigma}^{-1})} \right\| = 1$, which yields the setting in Feldman et al. [2014] of $c = \text{Tr}(\mathbf{P}) - 2\|\boldsymbol{\Sigma}^{-\frac{1}{2}} \mathbf{P} \boldsymbol{\Sigma}^{\frac{1}{2}}\| = d - 3$:

$$\hat{\boldsymbol{\mu}}^{\text{Bock}}(\mathbf{y}) = \hat{\boldsymbol{\mu}}^{\text{pooled}}(\mathbf{y}) + \left( 1 - \frac{d - 3}{\|\mathbf{y} - \hat{\boldsymbol{\mu}}^{\text{pooled}}(\mathbf{y})\|^2_{\boldsymbol{\Sigma}^{-1}}} \right)_+ (\mathbf{y} - \hat{\boldsymbol{\mu}}^{\text{pooled}}(\mathbf{y})) \tag{17}$$

Lastly, we note that it is straightforward to extend SureMap to the case of non-diagonal covariance matrices $\boldsymbol{\Sigma}$, i.e., when we want to simultaneously release performance estimates for groups with different numbers of intersections (e.g., just race, just age, and both race and age). In this case there will be nonzero covariance between elements of the vector $\mathbf{y}$, e.g., those corresponding to a specific age bracket and an intersection of that bracket with a specific race. To handle this, one only needs to derive the SURE objective estimating the (non-diagonal) $\boldsymbol{\Sigma}^{-1}$-weighted risk of the MAP estimator with (the same, non-diagonal) covariance $\boldsymbol{\Sigma}$; this can be done with the following generalization of Lemma B.1:

**Lemma B.2.** *Suppose* $\mathbf{y} \sim \mathcal{N}(\boldsymbol{\mu}, \boldsymbol{\Sigma})$ *for mean* $\boldsymbol{\mu} \in \mathbb{R}^d$ *and p.s.d. covariance* $\boldsymbol{\Sigma} \in \mathbb{R}^{d \times d}$, *and consider a function* $\hat{\boldsymbol{\mu}} : \mathbb{R}^d \to \mathbb{R}^d$ *s.t. for every* $i \in [d]$ *the function* $\hat{\mu}_i$ *is a.e. differentiable in* $y_i$ *and* $\mathbb{E}\|\boldsymbol{\nabla}_{\mathbf{y}} \cdot \hat{\boldsymbol{\mu}}(\mathbf{y})\|_1 < \infty$. *Then for any p.s.d.* $\mathbf{W} \in \mathbb{R}^{d \times d}$ *we have*

$$\mathbb{E}\|\hat{\boldsymbol{\mu}}(\mathbf{y}) - \boldsymbol{\mu}\|^2_{\mathbf{W}} = \mathbb{E}\|\hat{\boldsymbol{\mu}}(\mathbf{y}) - \mathbf{y}\|^2_{\mathbf{W}} + \text{Tr}(\mathbf{W}\boldsymbol{\Sigma}) + 2\mathbb{E}\sum_{i=1}^{d}\sum_{j=1}^{d}(\boldsymbol{\Sigma} \odot (\mathbf{W}\boldsymbol{\nabla}_{\mathbf{y}}\hat{\boldsymbol{\mu}}(\mathbf{y}) - \mathbf{W}))_{i,j} \tag{18}$$

*where* $\boldsymbol{\nabla}_{\mathbf{y}}\hat{\boldsymbol{\mu}}(\mathbf{y})$ *is the Jacobian of* $\hat{\boldsymbol{\mu}}$ *with entries* $\nabla_{\mathbf{y};i,j}\hat{\boldsymbol{\mu}}(\mathbf{y}) = \partial_{y_j}\hat{\mu}_i(\mathbf{y})$.

*Proof.* Note that since $\mathbb{E}\mathbf{y} = \boldsymbol{\mu}$ we have by a multivariate version of Stein's lemma (e.g., Liu [1994, Lemma 1]) that the following identity holds for any a.e. continuous $f : \mathbb{R}^d \to \mathbb{R}$ s.t. $\mathbb{E}|\partial_{y_i}f(\mathbf{y})| < \infty \ \forall \ i \in [d]$:

$$\mathbb{E}[(\mathbf{y} - \boldsymbol{\mu})_i f(\mathbf{y})] = \text{Cov}((\mathbf{y} - \boldsymbol{\mu})_i, f(\mathbf{y})) = \langle \boldsymbol{\Sigma}_{i,:}, \boldsymbol{\nabla}_{\mathbf{y}}f(\mathbf{y}) \rangle \tag{19}$$

Using this equality with $f(\mathbf{y}) = (\mathbf{W}(\hat{\boldsymbol{\mu}} - \mathbf{y}))_i$ yields

$$\begin{aligned}
\mathbb{E}&\|\hat{\boldsymbol{\mu}}(\mathbf{y}) - \boldsymbol{\mu}\|^2_{\mathbf{W}} \\
&= \mathbb{E}\|\hat{\boldsymbol{\mu}}(\mathbf{y}) - \mathbf{y}\|^2_{\mathbf{W}} + \mathbb{E}\|\mathbf{y} - \boldsymbol{\mu}\|^2_{\mathbf{W}} + 2\mathbb{E}\langle\hat{\boldsymbol{\mu}}(\mathbf{y}) - \mathbf{y}, \mathbf{W}(\mathbf{y} - \boldsymbol{\mu})\rangle \\
&= \mathbb{E}\|\hat{\boldsymbol{\mu}}(\mathbf{y}) - \mathbf{y}\|^2_{\mathbf{W}} + \text{Tr}(\mathbf{W}\boldsymbol{\Sigma}) + 2\mathbb{E}\sum_{i=1}^{d}\langle\boldsymbol{\Sigma}_{i,:}, \boldsymbol{\nabla}_{\mathbf{y}}((\mathbf{W}\hat{\boldsymbol{\mu}}(\mathbf{y}))_i - (\mathbf{W}\mathbf{y})_i)\rangle \\
&= \mathbb{E}\|\hat{\boldsymbol{\mu}}(\mathbf{y}) - \mathbf{y}\|^2_{\mathbf{W}} + \text{Tr}(\mathbf{W}\boldsymbol{\Sigma}) + 2\mathbb{E}\sum_{i=1}^{d}\sum_{j=1}^{d}\Sigma_{i,j}\partial_{y_j}(\langle\mathbf{W}_{i,:}, \hat{\boldsymbol{\mu}}(\mathbf{y})\rangle - \mathbf{W}_{i,:}\mathbf{y}) \quad (20) \\
&= \mathbb{E}\|\hat{\boldsymbol{\mu}}(\mathbf{y}) - \mathbf{y}\|^2_{\mathbf{W}} + \text{Tr}(\mathbf{W}\boldsymbol{\Sigma}) + 2\mathbb{E}\sum_{i=1}^{d}\sum_{j=1}^{d}\Sigma_{i,j}(\langle\mathbf{W}_{i,:}, \boldsymbol{\nabla}_{y_j}\hat{\boldsymbol{\mu}}(\mathbf{y})\rangle - W_{i,j}) \\
&= \mathbb{E}\|\hat{\boldsymbol{\mu}}(\mathbf{y}) - \mathbf{y}\|^2_{\mathbf{W}} + \text{Tr}(\mathbf{W}\boldsymbol{\Sigma}) + 2\mathbb{E}\sum_{i=1}^{d}\sum_{j=1}^{d}(\boldsymbol{\Sigma} \odot (\mathbf{W}\boldsymbol{\nabla}_{\mathbf{y}}\hat{\boldsymbol{\mu}}(\mathbf{y}) - \mathbf{W}))_{i,j}
\end{aligned}$$

$\square$

## C    SureMap estimator

In this section we describe the prior in full for any number of attributes, prove expressivity results reference in §3.1.2, and describe how to compute the estimator using coordinate descent.

### C.1    A linear prior

Suppose each group $g \in [d]$ corresponds to an intersection $\mathbf{g} \in \times_{a=1}^{k}[d_a]$ of $k$ attributes, with attribute $a$ having $d_a$ possible classes.[5] For example, the first attribute could be one of $d_1$ different age brackets and the second could be one of $d_2$ different racial categories. For each subset $A \subset [k]$ of attributes define a random tensor $\mathbf{Z}_A \in \mathbb{R}^{\times_{a \in A} d_a}$ with i.i.d. entries $Z_{A;\mathbf{c}} \sim \mathcal{N}(0,1)$, where $\mathbf{c} \in \times_{a \in A}[d_a]$ is a specific class combination of the attribute $A$. Then the **additive intersectional effects prior** on the mean $\mu_g$ of group $g$ is the weighted sum

$$\mu_g = \theta_g + \sum_{A \in 2^{[k]}} \tau_A Z_{A;\mathbf{g}_A} \tag{21}$$

with coefficients $\tau_A \in \mathbb{R}$ corresponding to each $A \in 2^{[k]}$. These hyperparameters thus determine the strength of the effect of attribute intersection $A$ on the group means, with $\tau_A Z_{A;\mathbf{c}}$ being added to the means of all groups $g$ s.t. $\mathbf{g}_A = \mathbf{c}$, i.e. whose attribute intersection results in the specific class combination $\mathbf{c}$.

Letting $\boldsymbol{\tau} = (\tau_\emptyset, \dots, \tau_{[k]}) \in \mathbb{R}^{2^k}_{\geq 0}$ be the vector of hyperparameters $\tau_A$ and assuming $\tau_{[k]} > 0$, this prior is equivalent to a Gaussian prior $\boldsymbol{\mu} \sim \mathcal{N}(\boldsymbol{\theta}, \boldsymbol{\Lambda}(\boldsymbol{\tau}))$ with covariance

$$\boldsymbol{\Lambda}(\boldsymbol{\tau}) = \sum_{A \in 2^{[k]}} \tau_A^2 \mathbf{U}_A \mathbf{U}_A^\top = \sum_{A \in 2^{[k]}} \tau_A^2 \mathbf{C}_A \tag{22}$$

for matrices $\mathbf{U}_A \in \{0,1\}^{d \times \prod_{a \in A} d_a}$ with orthogonal column vectors, each one corresponding to a different attribute intersection in $\times_{a \in A}[d_a]$ and its $g$th entry indicating whether group $g$ is a subset of that intersection. Their outer product matrices $\mathbf{C}_A = \mathbf{U}_A \mathbf{U}_A^\top$ have entries $C_{A;g,h}$ that are one if $\mathbf{g}_A = \mathbf{h}_A$ and zero otherwise, i.e. they indicate if groups $g$ and $h$ have the same type (e.g. (senior, female)) of attribute intersection $A$ (e.g. (age, sex)). In the simplest case, if we are disaggregating across just one attribute, i.e. $k = 1$ and $d_1 = d$, then $\boldsymbol{\Lambda}(\boldsymbol{\tau}) = \tau_\emptyset^2 \mathbf{1}_{d \times d} + \tau_{[k]}^2 \mathbf{I}_d$. Since $k \leq \lfloor \log_2 d \rfloor$, using this prior to define the covariance matrix reduces the number of hyperparameters to $d + 2^k \leq 2d$ for $\hat{\boldsymbol{\mu}}^{\mathrm{MAP}}_{\boldsymbol{\theta}, \boldsymbol{\Lambda}(\boldsymbol{\tau})}$, compared to $\mathcal{O}(d^2)$ for a general covariance $\boldsymbol{\Lambda}$.

### C.2    Expressivity of the linear prior

**Theorem C.1.** *Consider a disaggregated setting with $k$ attributes with $d_1, \dots, d_k$ possible categories, respectively, and total number of groups $d = \prod_{a=1}^{k} d_a$. If we use a diagonal covariance $\boldsymbol{\Sigma} \in \mathbb{R}^{d \times d}$ satisfying $\Sigma_{h,h} = \sigma^2 / n_h \; \forall \; h$ then the following holds $\forall \; \mathbf{y} \in \mathbb{R}^d$:*

*1. If $\tau_A = 0 \; \forall \; A \neq [k]$ and $\boldsymbol{\theta} \in \mathbb{R}^d$ then $\lim_{\tau_{[k]} \to \infty} \hat{\boldsymbol{\mu}}^{\mathrm{MAP}}_{\boldsymbol{\theta}, \boldsymbol{\Lambda}(\boldsymbol{\tau})}(\mathbf{y}) = \hat{\boldsymbol{\mu}}^{\mathrm{naive}}(\mathbf{y})$*

*2. If $\tau_A = 0 \; \forall \; A \notin \{\emptyset, [k]\}$ then $\lim_{\substack{\tau_\emptyset \to \infty \\ \tau_{[k]} \to 0}} \hat{\boldsymbol{\mu}}^{\mathrm{MAP}}_{\mathbf{0}_d, \boldsymbol{\Lambda}(\boldsymbol{\tau})}(\mathbf{y}) = \hat{\boldsymbol{\mu}}^{\mathrm{pooled}}(\mathbf{y})$*

*3. If $\tau_A = 0 \; \forall \; A \neq [k]$ and $\boldsymbol{\theta} \in \mathbb{R}^d$ then $\lim_{\tau_{[k]} \to 0} \hat{\boldsymbol{\mu}}^{\mathrm{MAP}}_{\boldsymbol{\theta}, \boldsymbol{\Lambda}(\boldsymbol{\tau})}(\mathbf{y}) = \boldsymbol{\theta}$*

*4. If $\tau_A = 0 \; \forall \; A \notin \{\emptyset, [k]\}$ and $\boldsymbol{\theta} \in \mathbb{R}^d$ then $\lim_{\substack{\tau_\emptyset \to \infty \\ \tau_{[k]} \to 0}} \hat{\boldsymbol{\mu}}^{\mathrm{MAP}}_{\boldsymbol{\theta}, \boldsymbol{\Lambda}(\boldsymbol{\tau})}(\mathbf{y}) = \hat{\boldsymbol{\mu}}^{\mathrm{pooled}}(\mathbf{y} - \boldsymbol{\theta}) + \boldsymbol{\theta}$*

*as desired.*

---

[5] Note that this does *not* reduce the generality of our basic setup, which can be recovered with $k = 1$ and $d_1 = d$.

*Proof.* The first and third results can be easily shown using the fact that $\boldsymbol{\Sigma}$ is diagonal:

$$\lim_{\tau_{[k]}\to\infty} \hat{\boldsymbol{\mu}}^{\mathrm{MAP}}_{\mathbf{0}_d,\boldsymbol{\Lambda}(\boldsymbol{\tau})}(\mathbf{y}) = \lim_{\tau_{[k]}\to\infty} \left(\frac{\mathbf{I}_d}{\tau_{[k]}^2} + \boldsymbol{\Sigma}^{-1}\right)^{-1}\left(\frac{\boldsymbol{\theta}}{\tau_{[k]}^2} + \boldsymbol{\Sigma}^{-1}\mathbf{y}\right) = \mathbf{y} = \hat{\boldsymbol{\mu}}^{\mathrm{naive}}(\mathbf{y}) \tag{23}$$

$$\lim_{\tau_{[k]}\to 0} \hat{\boldsymbol{\mu}}^{\mathrm{MAP}}_{\boldsymbol{\theta},\boldsymbol{\Lambda}(\boldsymbol{\tau})}(\mathbf{y}) = \lim_{\tau_{[k]}\to 0} \left(\frac{\mathbf{I}_d}{\tau_{[k]}^2} + \boldsymbol{\Sigma}^{-1}\right)^{-1}\left(\frac{\boldsymbol{\theta}}{\tau_{[k]}^2} + \boldsymbol{\Sigma}^{-1}\mathbf{y}\right) = \boldsymbol{\theta} \tag{24}$$

The second result follows from the last by substituting $\boldsymbol{\theta} = \mathbf{0}_d$, so we just need to prove the latter one. First note that

$$\hat{\boldsymbol{\mu}}^{\mathrm{MAP}}_{\boldsymbol{\theta},\boldsymbol{\Lambda}(\boldsymbol{\tau})}(\mathbf{y}) = \left(\boldsymbol{\Lambda}^{-1}(\boldsymbol{\tau}) + \boldsymbol{\Sigma}^{-1}\right)^{-1}\left(\boldsymbol{\Lambda}^{-1}(\boldsymbol{\tau})\boldsymbol{\theta} + \boldsymbol{\Sigma}^{-1}\mathbf{y}\right) \tag{25}$$

$$= \boldsymbol{\theta} + \left(\boldsymbol{\Lambda}^{-1}(\boldsymbol{\tau}) + \boldsymbol{\Sigma}^{-1}\right)^{-1}\boldsymbol{\Sigma}^{-1}(\mathbf{y} - \boldsymbol{\theta}) \tag{26}$$

so we just need to show that the second term approaches $\hat{\boldsymbol{\mu}}^{\mathrm{pooled}}(\mathbf{y}-\theta)$. We use the Sherman-Morrison formula to compute

$$\boldsymbol{\Lambda}^{-1}(\boldsymbol{\tau}) = (\tau_{[k]}^2\mathbf{I}_d + \tau_\emptyset^2\mathbf{1}_d\mathbf{1}_d^\top)^{-1} = \frac{\mathbf{I}_d}{\tau_{[k]}^2} - \frac{\tau_\emptyset^2\mathbf{1}_d\mathbf{1}_d^\top}{\tau_{[k]}^4 + \tau_{[k]}^2\tau_\emptyset^2 d} \tag{27}$$

and apply it again to compute

$$(\boldsymbol{\Lambda}^{-1}(\boldsymbol{\tau}) + \boldsymbol{\Sigma}^{-1})^{-1} = \left(\frac{\mathbf{I}_d}{\tau_{[k]}^2} - \frac{\tau_\emptyset^2\mathbf{1}_d\mathbf{1}_d^\top}{\tau_{[k]}^2 + \tau_\emptyset^2 d} + \boldsymbol{\Sigma}^{-1}\right)^{-1}$$

$$= \left(\frac{\mathbf{I}_d}{\tau_{[k]}^2} + \boldsymbol{\Sigma}^{-1}\right)^{-1} + \frac{\tau_\emptyset^2\left(\frac{\mathbf{I}_d}{\tau_{[k]}^2} + \boldsymbol{\Sigma}^{-1}\right)^{-1}\mathbf{1}_d\mathbf{1}_d^\top\left(\frac{\mathbf{I}_d}{\tau_{[k]}^2} + \boldsymbol{\Sigma}^{-1}\right)^{-1}}{\tau_{[k]}^4 + \tau_{[k]}^2\tau_\emptyset^2 d - \tau_\emptyset^2\,\mathrm{Tr}\left(\left(\frac{\mathbf{I}_d}{\tau_{[k]}^2} + \boldsymbol{\Sigma}^{-1}\right)^{-1}\right)} \tag{28}$$

Defining $\boldsymbol{\delta} = \mathbf{y} - \boldsymbol{\theta}$, we have by L'Hôpital's rule that $\forall\, g \in [d]$

$$\lim_{\substack{\tau_\emptyset\to\infty \\ \tau_{[k]}\to 0}}\left((\boldsymbol{\Lambda}^{-1}(\boldsymbol{\tau}) + \boldsymbol{\Sigma}^{-1})^{-1}\boldsymbol{\Sigma}^{-1}\boldsymbol{\delta}\right)_g$$

$$= \lim_{\substack{\tau_\emptyset\to\infty \\ \tau_{[k]}\to 0}} \frac{\delta_g/\Sigma_{g,g}}{\frac{1}{\tau_{[k]}^2} + \frac{1}{\Sigma_{g,g}}} + \frac{\frac{\tau_{[k]}^2\tau_\emptyset^2}{1+\frac{\tau_{[k]}^2}{\Sigma_{g,g}}}\sum_{h=1}^d \frac{\delta_h/\Sigma_{h,h}}{1+\frac{\tau_{[k]}^2}{\Sigma_{h,h}}}}{\tau_{[k]}^2 + \tau_\emptyset^2 d - \sum_{h=1}^d \frac{\tau_\emptyset^2}{1+\frac{\tau_{[k]}^2}{\Sigma_{h,h}}}}$$

$$= \lim_{\tau_\emptyset\to\infty} \frac{\lim_{\tau_{[k]}\to\infty} \frac{\tau_\emptyset^2}{1+\frac{\tau_{[k]}^2}{\Sigma_{g,g}}}\left(\sum_{h=1}^d \frac{\delta_h/\Sigma_{h,h}}{1+\frac{\tau_{[k]}^2}{\Sigma_{h,h}}} - \frac{\tau_{[k]}^2\delta_h/\Sigma_{h,h}^2}{\left(1+\frac{\tau_{[k]}^2}{\Sigma_{h,h}}\right)^2} - \frac{\tau_{[k]}^2/\Sigma_{g,g}}{\left(1+\frac{\tau_{[k]}^2}{\Sigma_{g,g}}\right)^2}\sum_{h=1}^d \frac{\delta_h/\Sigma_{h,h}}{1+\frac{\tau_{[k]}^2}{\Sigma_{h,h}}}\right)}{\lim_{\tau_{[k]}\to\infty} 1 + \tau_\emptyset^2\sum_{h=1}^d \frac{\Sigma_{h,h}}{\left(\tau_{[k]}^2+\Sigma_{h,h}\right)^2}} \tag{29}$$

$$= \lim_{\tau_\emptyset\to\infty} \frac{\tau_\emptyset^2\sum_{h=1}^d \frac{\delta_h}{\Sigma_{h,h}}}{1+\sum_{h=1}^d \frac{\tau_\emptyset^2}{\Sigma_{h,h}}} = \frac{1}{n}\sum_{h=1}^d n_h(y_h - \theta_h) = \hat{\mu}^{\mathrm{pooled}}_g(\mathbf{y} - \boldsymbol{\theta})$$

as desired. $\qquad\square$

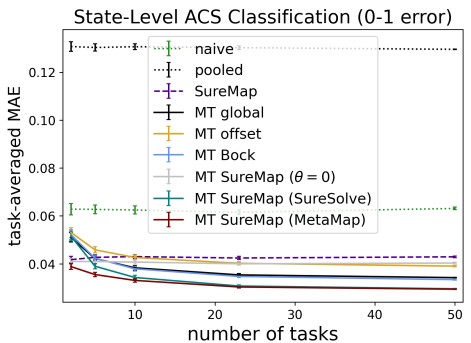 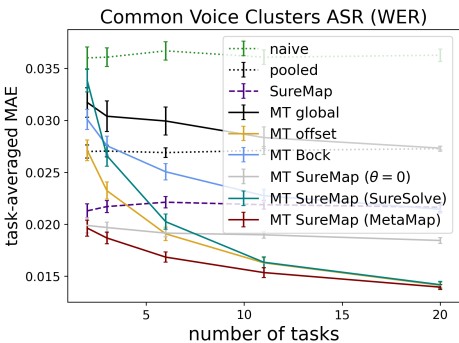

Figure 6: Performance as the number of tasks varies, evaluated on SLACS (left) and CVC (right). This plot is similar to Figure 5 but demonstrates the superiority of the multi-task version of SureMap that we choose (MetaMap) relative to alternatives.

### C.3  How to set the multi-task center

In the single-task case we set the prior mean $\boldsymbol{\theta} = \mathbf{0}_d$; this can also be done in the multi-task setting. Alternatively, we can use multi-task data to construct estimator $\hat{\boldsymbol{\theta}}$ of some true underlying multi-task mean $\boldsymbol{\theta}$ and substitute the former for $\boldsymbol{\theta}$; for simplicity we will restrict to the ourselves to *linear* estimators $\hat{\boldsymbol{\theta}} = \sum_{t=1}^T \mathbf{M}_t \mathbf{y}_t$, where the matrices $\mathbf{M}_1, \dots, \mathbf{M}_T$ are independent of $\mathbf{y}_1, \dots, \mathbf{y}_T$. To determine these matrices, we will assign them some parametric form and set those parameters by minimizing the sum of the $\boldsymbol{\Sigma}_t^{-1}$-weighted risks across tasks, as estimated by SURE:

$$
\sum_{t=1}^T \mathbb{E}_t \| \hat{\boldsymbol{\mu}}_{\hat{\boldsymbol{\theta}}, \boldsymbol{\Lambda}}^{\mathrm{MAP}}(\mathbf{y}_t) - \boldsymbol{\mu}_t \|_{\boldsymbol{\Sigma}_t^{-1}}^2 = \sum_{t=1}^T \mathbb{E}_t \left( \| \hat{\boldsymbol{\mu}}_{\hat{\boldsymbol{\theta}}, \boldsymbol{\Lambda}}^{\mathrm{MAP}}(\mathbf{y}_t) - \mathbf{y}_t \|_{\boldsymbol{\Sigma}_t^{-1}}^2 - d + 2 \boldsymbol{\nabla}_{\mathbf{y}_t} \cdot \hat{\boldsymbol{\mu}}_{\hat{\boldsymbol{\theta}}, \boldsymbol{\Lambda}}^{\mathrm{MAP}}(\mathbf{y}_t) \right)
$$

$$
= dT + \sum_{t=1}^T \mathbb{E}_t \| \mathbf{A}_t (\hat{\boldsymbol{\theta}} - \mathbf{y}_t) \|_{\boldsymbol{\Sigma}_t^{-1}}^2 + 2 \operatorname{Tr}(\mathbf{A}_t \mathbf{M}_t - \mathbf{A}_t)
$$

(30)

for $\mathbf{A}_t = (\boldsymbol{\Lambda}^{-1} + \boldsymbol{\Sigma}_t^{-1})^{-1} \boldsymbol{\Lambda}^{-1}$, i.e. $\hat{\boldsymbol{\mu}}_{\hat{\boldsymbol{\theta}}, \boldsymbol{\Lambda}}^{\mathrm{MAP}}(\mathbf{y}_t) = \mathbf{y}_t + \mathbf{A}_t (\hat{\boldsymbol{\theta}} - \mathbf{y}_t) = \mathbf{y}_t - \mathbf{A}_t (\mathbf{y}_t - \sum_{s=1}^T \mathbf{M}_s \mathbf{y}_s)$.

#### C.3.1  SureSolve

This approach sets the prior mean by finding the $\boldsymbol{\theta}$ that minimizes the above SURE objective, which is equivalent to solving an overconstrained linear system:

$$
\hat{\boldsymbol{\theta}} = dT + \underset{\boldsymbol{\theta} \in \mathbb{R}^d}{\arg\min} \sum_{t=1}^T \| \mathbf{A}_t (\boldsymbol{\theta} - \mathbf{y}_t) \|_{\boldsymbol{\Sigma}_t^{-1}}^2 - 2 \operatorname{Tr}(\mathbf{A}_t) = \left( \sum_{t=1}^T \mathbf{A}_t^\top \boldsymbol{\Sigma}_t^{-1} \mathbf{A}_t \right)^{-1} \sum_{t=1}^T \mathbf{A}_t^\top \boldsymbol{\Sigma}_t^{-1} \mathbf{A}_t \mathbf{y}_t
$$

(31)

Note that this estimator can be expressed in the desired linear form $\hat{\boldsymbol{\theta}} = \sum_{t=1}^T \mathbf{M}_t \mathbf{y}_t$, with the matrices $\mathbf{M}_t = \left( \sum_{s=1}^T \mathbf{A}_s^\top \boldsymbol{\Sigma}_s^{-1} \mathbf{A}_s \right)^{-1} \mathbf{A}_t^\top \boldsymbol{\Sigma}_t^{-1} \mathbf{A}_t$ depending on the parameters determining $\boldsymbol{\Lambda}$.

#### C.3.2  MetaMap

Alternatively, if we assume the prior is correct for some $\boldsymbol{\theta}$ and $\boldsymbol{\Lambda}$, i.e. $\mathbf{y}_t \sim \mathcal{N}(\boldsymbol{\theta}, \boldsymbol{\Lambda} + \boldsymbol{\Sigma}_t)$, then a natural estimator to use for $\boldsymbol{\theta}$ given a fixed $\boldsymbol{\Lambda}$ is a MAP estimator of $\hat{\boldsymbol{\theta}}_{\boldsymbol{\Gamma}}$ with prior $\mathcal{N}(\mathbf{0}_d, \boldsymbol{\Gamma})$ for some covariance $\boldsymbol{\Gamma}$. Define $\boldsymbol{\Sigma}_{\boldsymbol{\Lambda}} = (\sum_{t=1}^T (\boldsymbol{\Lambda} + \boldsymbol{\Sigma}_t)^{-1})^{-1}$ and note that $\mathbf{y}_{\boldsymbol{\Lambda}} = \boldsymbol{\Sigma}_{\boldsymbol{\Lambda}} \sum_{t=1}^T (\boldsymbol{\Lambda} + \boldsymbol{\Sigma}_t)^{-1} \mathbf{y}_t$ is distributed as $\mathcal{N}(\boldsymbol{\theta}, \boldsymbol{\Sigma}_{\boldsymbol{\Lambda}})$. Then the MAP estimator of $\boldsymbol{\theta}$

$$
\hat{\boldsymbol{\theta}}_{\boldsymbol{\Gamma}} = \left( \boldsymbol{\Gamma}^{-1} + \boldsymbol{\Sigma}_{\boldsymbol{\Lambda}}^{-1} \right)^{-1} \boldsymbol{\Sigma}_{\boldsymbol{\Lambda}}^{-1} \mathbf{y}_{\boldsymbol{\Lambda}} = \sum_{t=1}^T \left( \boldsymbol{\Gamma}^{-1} + \boldsymbol{\Sigma}_{\boldsymbol{\Lambda}}^{-1} \right)^{-1} (\boldsymbol{\Lambda} + \boldsymbol{\Sigma}_t)^{-1} \mathbf{y}_t
$$

(32)

has the necessary form $\hat{\boldsymbol{\theta}} = \sum_{t=1}^T \mathbf{M}_t \mathbf{y}_t$ for $\mathbf{M}_t = (\boldsymbol{\Gamma}^{-1} + \boldsymbol{\Sigma}_{\boldsymbol{\Lambda}}^{-1})^{-1} (\boldsymbol{\Lambda} + \boldsymbol{\Sigma}_t)^{-1}$. In this case the matrices depend on both $\boldsymbol{\Lambda}$ and $\boldsymbol{\Gamma}$, i.e. the covariance matrices of the prior *and* the meta-prior.

**Algorithm 2:** Multi-task SureMap.

---

**Input:** target $f : \mathcal{Z} \to \mathbb{R}$, samples $S_t \subset \mathcal{Z}$ for each task $t = 1, \ldots, T$, partition $\{\mathcal{Z}_g\}_{g=1}^d$ of $\mathcal{Z}$,
      each group $g$ an intersection of $k \in \mathbb{Z}_{>0}$ attributes

`// compute naive group means`
**for** *task* $t \in [T]$ **do**
    **for** *group* $g \in [d]$ **do**
        $n_{t;g} \leftarrow |S_t \cap \mathcal{Z}_g|$
        $y_{t;g} \leftarrow \frac{1}{n_{t;g}} \sum_{z \in S_t \cap \mathcal{Z}_g} f(z)$

`// estimate group variances`
$n = \sum_{t=1}^T |S_t|$
$\sigma^2 \leftarrow \frac{1}{n-dT} \sum_{t=1}^T \sum_{g=1}^d \sum_{z \in S_t \cap Z_g} (f(z) - y_{t;g})^2$
**for** *task* $t \in [T]$ **do**
    $\boldsymbol{\Sigma}_t^{-1} \leftarrow \mathrm{diag}(\mathbf{n}_t)/\sigma^2$

`// compute auxiliary matrices (avoids inverting` $\boldsymbol{\Lambda}$`)`
**Method** $\mathbf{A}_t(\boldsymbol{\tau})$**:**
    `// compute prior covariance (matrices` $\mathbf{C}_A$ `are defined in Appendix C.1)`
    $\boldsymbol{\Lambda} \leftarrow \sum_{A \in 2^{[k]}} \tau_A^2 \mathbf{C}_A$
    **Output:** $(\mathbf{I}_d + \boldsymbol{\Lambda}\boldsymbol{\Sigma}_t^{-1})^{-1}$

`// compute auxiliary matrices (avoids inverting` $\boldsymbol{\Sigma}_t^{-1}$ `and` $\boldsymbol{\Gamma}$`)`
**Method** $\mathbf{M}_t(\boldsymbol{\tau}, \boldsymbol{v})$**:**
    `// compute prior and meta-prior covariances`
    $\boldsymbol{\Lambda} \leftarrow \sum_{A \in 2^{[k]}} \tau_A^2 \mathbf{C}_A$
    $\boldsymbol{\Gamma} \leftarrow \sum_{A \in 2^{[k]}} v_A^2 \mathbf{C}_A$
    **Output:** $\left(\mathbf{I}_d + \boldsymbol{\Gamma}\sum_{s=1}^T \boldsymbol{\Sigma}_s^{-1}(\boldsymbol{\Lambda}\boldsymbol{\Sigma}_s^{-1} + \mathbf{I}_d)^{-1}\right)^{-1} \boldsymbol{\Gamma}\boldsymbol{\Sigma}_t^{-1}(\boldsymbol{\Lambda}\boldsymbol{\Sigma}_t^{-1} + \mathbf{I}_d)^{-1}$

`// estimates prior mean using MAP`
**Method** $\hat{\theta}(\boldsymbol{\tau}, \boldsymbol{v})$**:**
    **Output:** $\sum_{t=1}^T \mathbf{M}_t(\boldsymbol{\tau}, \boldsymbol{v})\mathbf{y}_t$

`// optimize the sum of SUREs across tasks using L-BFGS-B`
$\hat{\boldsymbol{\tau}}, \hat{\boldsymbol{v}} = \underset{\boldsymbol{\tau}, \boldsymbol{v} \in \mathbb{R}_{\geq 0}^{2^k}}{\arg\min} \sum_{t=1}^T \|\mathbf{A}_t(\boldsymbol{\tau})(\hat{\boldsymbol{\theta}}(\boldsymbol{\tau}, \boldsymbol{v}) - \mathbf{y}_t)\|_{\boldsymbol{\Sigma}_t^{-1}}^2 + 2\,\mathrm{Tr}\left(\mathbf{A}_t(\boldsymbol{\tau})(\mathbf{M}_t(\boldsymbol{\tau}, \boldsymbol{v}) - \mathbf{I}_d)\right)$

`// return an estimate of the group means of each task` $t$ `using MAP`
**for** *task* $t \in [T]$ **do**
    **Output:** $\mathbf{y}_t + \mathbf{A}_t(\hat{\boldsymbol{\tau}})(\hat{\boldsymbol{\theta}}(\hat{\boldsymbol{\tau}}, \hat{\boldsymbol{v}}) - \mathbf{y}_t)$

---

# D Computation

Here we note additional computational details.

## D.1 Nonnegativity of SureMap

In both the single and multi-task cases there is no guarantee that $\boldsymbol{\theta}$ is in the convex hull of the values in $\mathbf{y}$ or $\{\mathbf{y}_t\}_{t=1}^T$, and in fact it can be negative. Since the quantities we are estimating are usually nonnegative, in practice we do a post-hoc correction forcing $\boldsymbol{\theta}$ to be nonnegative.

## D.2 Optimization of SureMap

Both the single-task and multi-task variants of SureMap require solving optimization problems, the former over $\boldsymbol{\tau}^2 = (\tau_A^2)_{A \subseteq [k]}$ (Eq. 12) and the latter also over $\boldsymbol{v}^2 = (v_A^2)_{A \subseteq [k]}$ (Eq. 14), both of which are vectors in $\mathbb{R}_{\geq 0}^{2^k}$. We find that both problems can be quickly and efficiently optimized using L-BFGS-B over the entire domain $\mathbb{R}_{\geq 0}^{2^k}$ and $\mathbb{R}_{\geq 0}^{2 \cdot 2^k}$, respectively, using the default settings provided in

its `SciPy` implementation.[6] To initialize the algorithm we set all entries of $\boldsymbol{\tau}^2$ and $\boldsymbol{\upsilon}^2$ to zero except those corresponding to the entire set $[k]$, which we set to one.[7] In the remainder of this section, we describe how to compute gradients (passed to L-BFGS-B) of the multi-task SureMap objective

$$dT + \sum_{t=1}^{T} \mathbb{E}_t \left\| \mathbf{A}_t \left( \mathbf{y}_t - \sum_{s=1}^{T} \mathbf{M}_s \mathbf{y}_s \right) \right\|_{\boldsymbol{\Sigma}_t^{-1}}^2 + 2 \operatorname{Tr}(\mathbf{A}_t \mathbf{M}_t - \mathbf{A}_t) \tag{33}$$

where the matrices $\mathbf{M}_t$ differ based on whether we are using the SureSolve or MetaMap variant. The gradients are taken w.r.t. the tuning parameters, which are the coefficients $\tau_1^2, \ldots, \tau_{2^k}^2$ used to define the prior covariance $\boldsymbol{\Lambda} = \sum_{i=1}^{2^k} \tau_i^2 \mathbf{U}_i \mathbf{U}_i^\top$ and, in the second case, the coefficients $\upsilon_1^2, \ldots, \upsilon_{2^k}^2$ used to define the meta-prior covariance $\boldsymbol{\Gamma} = \sum_{i=1}^{2^k} \upsilon_i^2 \mathbf{U}_i \mathbf{U}_i^\top$.[8] Noting that $\mathbf{A}_t = (\boldsymbol{\Lambda} + \boldsymbol{\Sigma}_t^{-1})^{-1} \boldsymbol{\Lambda}^{-1} = (\mathbf{I}_d + \boldsymbol{\Lambda} \boldsymbol{\Sigma}_t^{-1})^{-1}$ and $(\boldsymbol{\Lambda} + \boldsymbol{\Sigma}_t)^{-1} = \boldsymbol{\Lambda}^{-1}(\mathbf{I}_d + \boldsymbol{\Sigma}_t \boldsymbol{\Lambda}^{-1})^{-1} = \boldsymbol{\Sigma}_t^{-1}(\mathbf{I}_d + \boldsymbol{\Lambda} \boldsymbol{\Sigma}_t^{-1})^{-1} = \boldsymbol{\Sigma}_t^{-1} \mathbf{A}_t$, we have the derivative $\partial_i \mathbf{A}_t = -\mathbf{A}_t \mathbf{U}_i \mathbf{U}_i^\top \boldsymbol{\Sigma}_t^{-1} \mathbf{A}_t$ w.r.t. $\tau_i^2$, which yields $\partial_i \operatorname{Tr}(\mathbf{A}_t) = \operatorname{Tr}(-\mathbf{A}_t \mathbf{U}_i \mathbf{U}_i^\top \boldsymbol{\Sigma}_t^{-1} \mathbf{A}_t)$ and

$$\begin{aligned}
\partial_i \|\mathbf{A}_t(\boldsymbol{\theta} - \mathbf{y}_t)\|_{\boldsymbol{\Sigma}_t^{-1}}^2 &= 2(\boldsymbol{\theta} - \mathbf{y}_t)^\top \mathbf{A}_t^\top \boldsymbol{\Sigma}_t^{-1} \partial_i \mathbf{A}_t (\boldsymbol{\theta} - \mathbf{y}_t) \\
&= -2(\boldsymbol{\theta} - \mathbf{y}_t)^\top \mathbf{A}_t^\top \boldsymbol{\Sigma}_t^{-1} \mathbf{A}_t \mathbf{U}_i \mathbf{U}_i^\top \boldsymbol{\Sigma}_t^{-1} \mathbf{A}_t (\boldsymbol{\theta} - \mathbf{y}_t)
\end{aligned} \tag{34}$$

### D.2.1 SureSolve

First note that

$$\begin{aligned}
\partial_i [\mathbf{A}_t^\top \boldsymbol{\Sigma}_t^{-1} \mathbf{A}_t] = \partial_i [\boldsymbol{\Sigma}_t^{-1} \mathbf{A}_t^2] &= \boldsymbol{\Sigma}_t^{-1}(\mathbf{A}_t \partial_i \mathbf{A}_t + \partial_i \mathbf{A}_t \mathbf{A}_t) \\
&= -\boldsymbol{\Sigma}_t^{-1}(\mathbf{A}_t^2 + \mathbf{A}_t) \mathbf{U}_i \mathbf{U}_i^\top \boldsymbol{\Sigma}_t^{-1}(\mathbf{A}_t^2 + \mathbf{A}_t)
\end{aligned} \tag{35}$$

where the first step follows by $\boldsymbol{\Sigma}_t^{-1} \mathbf{A}_t = \boldsymbol{\Sigma}_t^{-1}(\mathbf{I}_d + \boldsymbol{\Lambda} \boldsymbol{\Sigma}_t^{-1})^{-1} = (\mathbf{I}_d + \boldsymbol{\Sigma}_t^{-1} \boldsymbol{\Lambda})^{-1} \boldsymbol{\Sigma}_t^{-1} = \mathbf{A}_t^\top \boldsymbol{\Sigma}_t^{-1}$. Then for $\mathbf{M}_t = \left( \sum_{s=1}^{T} \mathbf{A}_s^\top \boldsymbol{\Sigma}_s^{-1} \mathbf{A}_s \right)^{-1} \mathbf{A}_t^\top \boldsymbol{\Sigma}_t^{-1} \mathbf{A}_t$ we have

$$\begin{aligned}
\partial_i \mathbf{M}_t &= \left( \sum_{s=1}^{T} \mathbf{A}_s^\top \boldsymbol{\Sigma}_s^{-1} \mathbf{A}_s \right)^{-1} \partial_i [\mathbf{A}_t^\top \boldsymbol{\Sigma}_t^{-1} \mathbf{A}_t] - \left( \sum_{s=1}^{T} \mathbf{A}_s^\top \boldsymbol{\Sigma}_s^{-1} \mathbf{A}_s \right)^{-1} \sum_{s=1}^{T} \partial_i [\mathbf{A}_s^\top \boldsymbol{\Sigma}_s^{-1} \mathbf{A}_s] \mathbf{M}_t \\
&= -2 \left( \sum_{s=1}^{T} \mathbf{A}_s^\top \boldsymbol{\Sigma}_s^{-1} \mathbf{A}_s \right)^{-1} \boldsymbol{\Sigma}_t^{-1}(\mathbf{A}_t^2 + \mathbf{A}_t) \mathbf{U}_i \mathbf{U}_i^\top \boldsymbol{\Sigma}_t^{-1}(\mathbf{A}_t^2 + \mathbf{A}_t) \\
&\quad + 2 \left( \sum_{s=1}^{T} \mathbf{A}_s^\top \boldsymbol{\Sigma}_s^{-1} \mathbf{A}_s \right)^{-1} \sum_{s=1}^{T} \boldsymbol{\Sigma}_s^{-1}(\mathbf{A}_s^2 + \mathbf{A}_s) \mathbf{U}_i \mathbf{U}_i^\top \boldsymbol{\Sigma}_s^{-1}(\mathbf{A}_s^2 + \mathbf{A}_s) \mathbf{M}_t \\
&= -2\mathbf{B}_{i,t} + 2 \sum_{s=1}^{T} \mathbf{B}_{i,s} \mathbf{M}_t
\end{aligned} \tag{36}$$

where $\mathbf{B}_{i,t} = \left( \sum_{s=1}^{T} \mathbf{A}_s^\top \boldsymbol{\Sigma}_s^{-1} \mathbf{A}_s \right)^{-1} \boldsymbol{\Sigma}_t^{-1}(\mathbf{A}_t^2 + \mathbf{A}_t) \mathbf{U}_i \mathbf{U}_i^\top \boldsymbol{\Sigma}_t^{-1}(\mathbf{A}_t^2 + \mathbf{A}_t)$. We can then compute

$$\begin{aligned}
\partial_i \operatorname{Tr}(\mathbf{A}_t \mathbf{M}_t) &= -2 \operatorname{Tr} \left( \mathbf{A}_t \left( \mathbf{B}_{i,t} - \sum_{s=1}^{T} \mathbf{B}_{i,s} \mathbf{M}_t \right) \right) - \operatorname{Tr}(\mathbf{A}_t \mathbf{U}_i \mathbf{U}_i^\top \boldsymbol{\Sigma}_t^{-1} \mathbf{A}_t \mathbf{M}_t) \\
&= -2 \operatorname{Tr} \left( \mathbf{A}_t \left( \mathbf{B}_{i,t} + \left( \mathbf{U}_i \mathbf{U}_i^\top \boldsymbol{\Sigma}_t^{-1} \mathbf{A}_t - \sum_{s=1}^{T} \mathbf{B}_{i,s} \right) \mathbf{M}_t \right) \right)
\end{aligned} \tag{37}$$

---

[6] `https://docs.scipy.org/doc/scipy/reference/optimize.minimize-lbfgsb.html`
[7] This initialization corresponds to setting the prior covariance to be the $d \times d$ identity.
[8] We use indices $i$ instead of subsets $A$ here for simplicity.

and

$$\partial_i \left\| \mathbf{A}_t \left( \mathbf{y}_t - \sum_{s=1}^{T} \mathbf{M}_s \mathbf{y}_s \right) \right\|_{\mathbf{\Sigma}_t^{-1}}^2 = \partial_i \sum_{s=1}^{T} \sum_{r=1}^{T} (\mathbf{y}_t - \mathbf{M}_s \mathbf{y}_s)^\top \mathbf{A}_t^\top \mathbf{\Sigma}_t^{-1} \mathbf{A}_t (\mathbf{y}_t - \mathbf{M}_r \mathbf{y}_r)$$

$$= -2 \sum_{s=1}^{T} (\partial_i \mathbf{M}_s \mathbf{y}_s)^\top \mathbf{A}_t^\top \mathbf{\Sigma}_t^{-1} \mathbf{A}_t (\mathbf{y}_t - \hat{\boldsymbol{\theta}})$$

$$+ 2(\mathbf{y}_t - \hat{\boldsymbol{\theta}})^\top \mathbf{A}_t^\top \mathbf{\Sigma}_t^{-1} \partial_i \mathbf{A}_t (\mathbf{y}_t - \hat{\boldsymbol{\theta}})$$

$$= 4 \sum_{s=1}^{T} \mathbf{y}_s^\top \mathbf{B}_{i,s}^\top \mathbf{A}_t^\top \mathbf{\Sigma}_t^{-1} \mathbf{A}_t (\mathbf{y}_t - \hat{\boldsymbol{\theta}}) \tag{38}$$

$$- 4\hat{\boldsymbol{\theta}}^\top \sum_{s=1}^{T} \mathbf{B}_{i,s}^\top \mathbf{A}_t^\top \mathbf{\Sigma}_t^{-1} \mathbf{A}_t (\mathbf{y}_t - \hat{\boldsymbol{\theta}})$$

$$- 2(\mathbf{y}_t - \hat{\boldsymbol{\theta}})^\top \mathbf{A}_t^\top \mathbf{\Sigma}_t^{-1} \mathbf{A}_t \mathbf{U}_i \mathbf{U}_i^\top \mathbf{\Sigma}_t^{-1} \mathbf{A}_t (\mathbf{y}_t - \hat{\boldsymbol{\theta}})$$

$$= 4 \sum_{s=1}^{T} (\mathbf{y}_s - \hat{\boldsymbol{\theta}})^\top \mathbf{B}_{i,s}^\top \mathbf{A}_t^\top \mathbf{\Sigma}_t^{-1} \mathbf{A}_t (\mathbf{y}_t - \hat{\boldsymbol{\theta}})$$

$$- 2(\mathbf{y}_t - \hat{\boldsymbol{\theta}})^\top \mathbf{A}_t^\top \mathbf{\Sigma}_t^{-1} \mathbf{A}_t \mathbf{U}_i \mathbf{U}_i^\top \mathbf{\Sigma}_t^{-1} \mathbf{A}_t (\mathbf{y}_t - \hat{\boldsymbol{\theta}})$$

### D.3 MetaMap

Note that $(\mathbf{\Lambda} + \mathbf{\Sigma}_t)^{-1} = \mathbf{\Sigma}_t^{-1}(\mathbf{\Lambda}\mathbf{\Sigma}_t^{-1} + \mathbf{I}_d) = \mathbf{\Sigma}_t^{-1}\mathbf{A}_t$ so we can write the matrices representing the estimator as $\mathbf{M}_t = (\mathbf{\Gamma}^{-1} + \mathbf{\Sigma}_{\mathbf{\Lambda}}^{-1})^{-1}(\mathbf{\Lambda} + \mathbf{\Sigma}_t)^{-1} = (\mathbf{\Gamma}^{-1} + \sum_{s=1}^{T} \mathbf{\Sigma}_s^{-1}\mathbf{A}_s)^{-1}\mathbf{\Sigma}_t^{-1}\mathbf{A}_t$. Thus

$$\partial_i \mathbf{M}_t = (\mathbf{\Gamma}^{-1} + \mathbf{\Sigma}_{\mathbf{\Lambda}}^{-1})^{-1}\mathbf{\Sigma}_t^{-1}\partial_i\mathbf{A}_t - (\mathbf{\Gamma}^{-1} + \mathbf{\Sigma}_{\mathbf{\Lambda}}^{-1})^{-1}\sum_{s=1}^{T}\mathbf{\Sigma}_s^{-1}\partial_i\mathbf{A}_s\mathbf{M}_t$$

$$= -\mathbf{M}_t\mathbf{U}_i\mathbf{U}_i^\top\mathbf{\Sigma}_t^{-1}\mathbf{A}_t + \sum_{s=1}^{T}\mathbf{M}_s\mathbf{U}_i\mathbf{U}_i^\top\mathbf{\Sigma}_s^{-1}\mathbf{A}_s\mathbf{M}_t \tag{39}$$

We can then compute

$$\partial_i \operatorname{Tr}(\mathbf{A}_t\mathbf{M}_t) = -\operatorname{Tr}(\mathbf{A}_t(\mathbf{M}_t\mathbf{A}_t + \mathbf{A}_t\mathbf{M}_t)\mathbf{U}_i\mathbf{U}_i^\top\mathbf{\Sigma}_t^{-1}) + \sum_{s=1}^{T}\operatorname{Tr}(\mathbf{A}_t\mathbf{M}_s\mathbf{U}_i\mathbf{U}_i^\top\mathbf{\Sigma}_s^{-1}\mathbf{A}_s\mathbf{M}_t) \tag{40}$$

and

$$\partial_i \left\| \mathbf{A}_t \left( \mathbf{y}_t - \sum_{s=1}^{T} \mathbf{M}_s \mathbf{y}_s \right) \right\|_{\mathbf{\Sigma}_t^{-1}}^2 = \partial_i \sum_{s=1}^{T} \sum_{r=1}^{T} (\mathbf{y}_t - \mathbf{M}_s \mathbf{y}_s)^\top \mathbf{A}_t^\top \mathbf{\Sigma}_t^{-1} \mathbf{A}_t (\mathbf{y}_t - \mathbf{M}_r \mathbf{y}_r)$$

$$= -2 \sum_{s=1}^{T} (\partial_i \mathbf{M}_s \mathbf{y}_s)^\top \mathbf{A}_t^\top \mathbf{\Sigma}_t^{-1} \mathbf{A}_t (\mathbf{y}_t - \hat{\boldsymbol{\theta}}_{\mathbf{\Gamma}})$$

$$+ 2(\mathbf{y}_t - \hat{\boldsymbol{\theta}}_{\mathbf{\Gamma}})^\top \mathbf{A}_t^\top \mathbf{\Sigma}_t^{-1} \partial_i \mathbf{A}_t (\mathbf{y}_t - \hat{\boldsymbol{\theta}}_{\mathbf{\Gamma}})$$

$$= 2 \sum_{s=1}^{T} \mathbf{y}_s^\top (\mathbf{M}_s \mathbf{U}_i \mathbf{U}_i^\top \mathbf{\Sigma}_s^{-1} \mathbf{A}_s)^\top \mathbf{A}_t \mathbf{\Sigma}_t^{-1} \mathbf{A}_t (\mathbf{y}_t - \hat{\boldsymbol{\theta}}_{\mathbf{\Gamma}})$$

$$- 2 \sum_{s=1}^{T} \sum_{r=1}^{T} \mathbf{y}_s^\top (\mathbf{M}_r \mathbf{U}_i \mathbf{U}_i^\top \mathbf{\Sigma}_r^{-1} \mathbf{A}_r \mathbf{M}_s)^\top \mathbf{A}_t \mathbf{\Sigma}_t^{-1} \mathbf{A}_t (\mathbf{y}_t - \hat{\boldsymbol{\theta}}_{\mathbf{\Gamma}})$$

$$- 2(\mathbf{y}_t - \hat{\boldsymbol{\theta}}_{\mathbf{\Gamma}})^\top \mathbf{A}_t^\top \mathbf{\Sigma}_t^{-1} \mathbf{A}_t \mathbf{U}_i \mathbf{U}_i^\top \mathbf{\Sigma}_t^{-1} \mathbf{A}_t (\mathbf{y}_t - \hat{\boldsymbol{\theta}}_{\mathbf{\Gamma}})$$

$$= 2 \sum_{s=1}^{T} (\mathbf{y}_s - \hat{\boldsymbol{\theta}}_{\mathbf{\Gamma}})^\top \mathbf{A}_s^\top \mathbf{\Sigma}_s^{-1} \mathbf{U}_i \mathbf{U}_i^\top \mathbf{M}_s^\top \mathbf{A}_t^\top \mathbf{\Sigma}_t^{-1} \mathbf{A}_t (\mathbf{y}_t - \hat{\boldsymbol{\theta}}_{\mathbf{\Gamma}})$$

$$- 2(\mathbf{y}_t - \hat{\boldsymbol{\theta}}_{\mathbf{\Gamma}})^\top \mathbf{A}_t^\top \mathbf{\Sigma}_t^{-1} \mathbf{A}_t \mathbf{U}_i \mathbf{U}_i^\top \mathbf{\Sigma}_t^{-1} \mathbf{A}_t (\mathbf{y}_t - \hat{\boldsymbol{\theta}}_{\mathbf{\Gamma}})$$

(41)

Lastly, note that $\mathbf{M}_t = (\mathbf{\Gamma}^{-1} + \mathbf{\Sigma}_{\mathbf{\Lambda}}^{-1})^{-1} \mathbf{\Sigma}_t^{-1} \mathbf{A}_t = (\mathbf{I}_d + \mathbf{\Gamma} \mathbf{\Sigma}_{\mathbf{\Lambda}}^{-1})^{-1} \mathbf{\Gamma} \mathbf{\Sigma}_t^{-1} \mathbf{A}_t$ and redefine the derivative $\partial_i$ to be w.r.t. $v_i^2$, so that

$$\partial_i \mathbf{M}_t = -(\mathbf{I}_d + \mathbf{\Gamma} \mathbf{\Sigma}_{\mathbf{\Lambda}}^{-1})^{-1} \mathbf{U}_i \mathbf{U}_i^\top \mathbf{\Sigma}_{\mathbf{\Lambda}}^{-1} \mathbf{M}_t + (\mathbf{I}_d + \mathbf{\Gamma} \mathbf{\Sigma}_{\mathbf{\Lambda}}^{-1})^{-1} \mathbf{U}_i \mathbf{U}_i^\top \mathbf{\Sigma}_t^{-1} \mathbf{A}_t \qquad (42)$$

Then we have $\partial_i \operatorname{Tr}(\mathbf{A}_t \mathbf{M}_t) = \operatorname{Tr}(\mathbf{A}_t (\mathbf{I}_d + \mathbf{\Gamma} \mathbf{\Sigma}_{\mathbf{\Lambda}}^{-1})^{-1} \mathbf{U}_i \mathbf{U}_i^\top (\mathbf{\Sigma}_t^{-1} \mathbf{A}_t - \mathbf{\Sigma}_{\mathbf{\Lambda}}^{-1} \mathbf{M}_t))$ and

$$\partial_i \left\| \mathbf{A}_t \left( \mathbf{y}_t - \hat{\boldsymbol{\theta}}_{\mathbf{\Gamma}} \right) \right\|_{\mathbf{\Sigma}_t^{-1}}^2$$

$$= -2 \sum_{s=1}^{T} (\partial_i \mathbf{M}_s \mathbf{y}_s)^\top \mathbf{A}_t^\top \mathbf{\Sigma}_t^{-1} \mathbf{A}_t (\mathbf{y}_t - \hat{\boldsymbol{\theta}}_{\mathbf{\Gamma}})$$

$$= 2 \left( \mathbf{\Sigma}_{\mathbf{\Lambda}}^{-1} \hat{\boldsymbol{\theta}}_{\mathbf{\Gamma}} - \sum_{s=1}^{T} \mathbf{\Sigma}_s^{-1} \mathbf{A}_s \mathbf{y}_s \right)^\top \mathbf{U}_i \mathbf{U}_i^\top (\mathbf{I}_d + \mathbf{\Gamma} \mathbf{\Sigma}_{\mathbf{\Lambda}}^{-1})^{-\top} \mathbf{A}_t^\top \mathbf{\Sigma}_t^{-1} \mathbf{A}_t (\mathbf{y}_t - \hat{\boldsymbol{\theta}}_{\mathbf{\Gamma}})$$

(43)

### D.4 Handling groups with no data

It is frequently the case that a specific group $g$ may not have any examples, i.e., $n_g = 0$, and so we cannot define $\Sigma_{g,g} = \sigma^2 / n_g$. At the same time, we may need to handle the dimension associated with this group, either because we still need to report a value for it or because we are in the multi-task setting and other tasks do have examples of that group that may allow us to get an estimate. To handle this issue, in all computations we only use the precision matrix $\mathbf{\Sigma}^{-1}$, which can be easily defined in the case of $n_g = 0$ using $\Sigma_{g,g}^{-1} = n_g / \sigma^2 = 0$. Note in particular that $(\mathbf{\Lambda} + \mathbf{\Sigma})^{-1} = \mathbf{\Sigma}^{-1} (\mathbf{\Lambda} \mathbf{\Sigma}^{-1} + \mathbf{I}_d)^{-1}$.

### D.5 Handling near-singular covariances

Since we would like to optimize over the entire domain $\boldsymbol{\tau}^2 \in \mathbb{R}_{\geq 0}^{2^k}$ but $\mathbf{\Lambda}(\mathbf{0}_{2^k}) = \mathbf{0}_{d \times d}$ is singular, we avoid inverting prior covariance matrices (i.e. $\mathbf{\Lambda}, \mathbf{\Gamma}$) in all computations. Note in particular that $\mathbf{A}_t = (\mathbf{\Lambda}^{-1} + \mathbf{\Sigma}^{-1})^{-1} \mathbf{\Lambda}^{-1} = (\mathbf{I}_d + \mathbf{\Lambda} \mathbf{\Sigma}^{-1})^{-1}$.

### D.6 Handling group variances for AUC

While obtaining unbiased variance estimates for averages is straightforward, it is more involved for multi-point statistics such as AUC. For simplicity we just use $(n_g + 1)/(12 n_g n_g^{(0)} n_g^{(1)})$, where $n_g^{(i)}$ is the number of members of group $g$ with label $i$; this estimate is derived from the variance estimate of the related Mann-Whitney $U$-statistic [Siegel, 1956]. However, future work may consider improvements based on more complicated approaches [Cortes and Mohri, 2003, Wang and Guo, 2020], or using bootstrapping techniques as is done by Herlihy et al. [2024] for structured regression.

## E Derivation of SureMap estimators via ridge regression

In this appendix we show that $\hat{\boldsymbol{\mu}}^{\mathrm{SM}}$ and $\hat{\boldsymbol{\mu}}_t^{\mathrm{SM}}$ can be derived as ridge regression estimators.

Similar to the notation in Appendix C, we consider $k$ sensitive attributes (like sex, age, etc.) indexed by $a \in [k]$. The $a$th sensitive attribute is denoted $g_a$ and takes values in $[d_a]$. The joint sensitive attribute $\mathbf{g}$ takes values in $\mathcal{G} = [d_1] \times [d_2] \times \cdots \times [d_k]$ with the cardinality $d = |\mathcal{G}| = d_1 d_2 \cdots d_k$. For $A \subseteq [k]$, write $\mathbf{g}_A$ for the tuple $(g_a)_{a \in A}$.

### E.1 Single-task regression model

Each joint attribute $\mathbf{g} \in \mathcal{G}$ identifies an intersectional group (of order $k$). We seek to jointly fit means of all these groups, represented as a vector $\boldsymbol{\mu} \in \mathbb{R}^{|\mathcal{G}|}$, based on the vector of empirical means $\mathbf{y} \in \mathbb{R}^{|\mathcal{G}|}$. We posit a regression model

$$\boldsymbol{\mu} = \boldsymbol{\Phi}\boldsymbol{\beta}$$

where $\boldsymbol{\Phi} \in \mathbb{R}^{|\mathcal{G}| \times |\mathcal{J}|}$ is a feature matrix and $\boldsymbol{\beta} \in \mathbb{R}^{|\mathcal{J}|}$ is a vector of regression coefficients. The columns of $\boldsymbol{\Phi}$ are referred to as features and indexed by $j \in \mathcal{J}$, where $\mathcal{J}$ is some index set. We assume a Gaussian prior on the parameter $\boldsymbol{\beta} \sim \mathcal{N}(\mathbf{0}, \mathbf{K})$ and a Gaussian distribution over observation errors, so $\mathbf{y} \sim \mathcal{N}(\boldsymbol{\Phi}\boldsymbol{\beta}, \boldsymbol{\Sigma})$, where $\mathbf{K}$ and $\boldsymbol{\Sigma}$ are, respectively, the prior covariance matrix and error covariance matrix.

The error covariance matrix $\boldsymbol{\Sigma}$ is assumed fixed and positive definite, the prior covariance matrix $\mathbf{K}$ is viewed as a hyperparameter (with a specific structure to reduce its dimension); for simplicity, we assume that $\mathbf{K}$ is positive definite (but that assumption is not necessary). Any generic forms of $\boldsymbol{\Phi}$, $\mathbf{K}$ and $\boldsymbol{\Sigma}$ can be considered. Here we exhibit a specific form of $\boldsymbol{\Phi}$ and $\mathbf{K}$ under which the MAP regression estimator is the same as $\hat{\boldsymbol{\mu}}^{\mathrm{SM}}$, when provided with the same error covariance matrix $\boldsymbol{\Sigma}$.

We consider a structured form of matrix $\boldsymbol{\Phi}$, with features being indicators of tuples of sensitive attribute values. The features are indexed by $j \in \mathcal{J}$, where

$$\mathcal{J} = \big\{(A, \mathbf{c}) : A \subseteq [k], \mathbf{c} \in \textstyle\prod_{a \in A}[d_a]\big\},$$

and defined as

$$\Phi_{\mathbf{g},(A,\mathbf{c})} = 1\{\mathbf{g}_A = \mathbf{c}\}. \tag{44}$$

We will also consider subsets of features that focus on a specific subset of sensitive attributes $A \subseteq [k]$,

$$\mathcal{J}_A = \big\{(A, \mathbf{c}) : \mathbf{c} \in \textstyle\prod_{a \in A}[d_a]\big\},$$

and write $\boldsymbol{\Phi}_A \in \mathbb{R}^{|\mathcal{G}| \times |\mathcal{J}_A|}$ for the submatrix composed of features indexed by $j \in \mathcal{J}_A$.

The prior matrix $\mathbf{K}$ is diagonal, specified using a set of hyperparameters $\boldsymbol{\tau} = (\tau_A)_{A \subseteq [k]}$, with diagonal entries

$$K_{(A,\mathbf{c}),(A,\mathbf{c})} = \tau_A^2. \tag{45}$$

The MAP solution under the model described above is obtained by solving

$$\hat{\boldsymbol{\beta}} = \arg\min_{\boldsymbol{\beta}} \Big[(\mathbf{y} - \boldsymbol{\Phi}\boldsymbol{\beta})^\top \boldsymbol{\Sigma}^{-1}(\mathbf{y} - \boldsymbol{\Phi}\boldsymbol{\beta}) + \boldsymbol{\beta}^\top \mathbf{K}^{-1}\boldsymbol{\beta}\Big],$$

which is a weighted ridge regression problem. Setting the gradient to zero, the solution must satisfy

$$-2\boldsymbol{\Phi}^\top \boldsymbol{\Sigma}^{-1}(\mathbf{y} - \boldsymbol{\Phi}\boldsymbol{\beta}) + 2\mathbf{K}^{-1}\boldsymbol{\beta} = \mathbf{0}.$$

Hence,

$$\hat{\boldsymbol{\beta}} = (\boldsymbol{\Phi}^\top \boldsymbol{\Sigma}^{-1} \boldsymbol{\Phi} + \mathbf{K}^{-1})^{-1} \boldsymbol{\Phi}^\top \boldsymbol{\Sigma}^{-1} \mathbf{y}.$$

This yields a regression-based estimator

$$\hat{\boldsymbol{\mu}}^{\text{reg}} = \boldsymbol{\Phi}\hat{\boldsymbol{\beta}} = \boldsymbol{\Phi}(\boldsymbol{\Phi}^\top \boldsymbol{\Sigma}^{-1} \boldsymbol{\Phi} + \mathbf{K}^{-1})^{-1} \boldsymbol{\Phi}^\top \boldsymbol{\Sigma}^{-1} \mathbf{y}. \tag{46}$$

In contrast, by Eqs. (11) and (7), the $\hat{\boldsymbol{\mu}}^{\text{SM}}$ estimator is obtained as

$$\hat{\boldsymbol{\mu}}^{\text{SM}} = (\boldsymbol{\Lambda}^{-1} + \boldsymbol{\Sigma}^{-1})^{-1} \boldsymbol{\Sigma}^{-1} \mathbf{y},$$

where the matrix $\boldsymbol{\Lambda}$ (following Eq. 22 in Appendix C.1) depends on the hyperparameter vector $\boldsymbol{\tau}$ as

$$\boldsymbol{\Lambda} = \sum_{A \subseteq [k]} \tau_A^2 \mathbf{U}_A \mathbf{U}_A^\top,$$

where $\mathbf{U}_A$ is precisely the submatrix $\boldsymbol{\Phi}_A$ defined above. Thus, writing $\phi_{A,\mathbf{c}} \in \mathbb{R}^{|\mathcal{G}|}$ for the column of $\boldsymbol{\Phi}$ indexed by $(A, \mathbf{c})$, we obtain

$$\boldsymbol{\Lambda} = \sum_{A \subseteq [k]} \tau_A^2 \boldsymbol{\Phi}_A \boldsymbol{\Phi}_A^\top = \sum_{A \subseteq [k]} \sum_{\mathbf{c} \in \prod_{a \in A}[d_a]} \tau_A^2 \phi_{A,\mathbf{c}} \phi_{A,\mathbf{c}}^\top = \boldsymbol{\Phi}\mathbf{K}\boldsymbol{\Phi}^\top.$$

Hence, the $\hat{\boldsymbol{\mu}}^{\text{SM}}$ estimator can be rewritten as

$$\hat{\boldsymbol{\mu}}^{\text{SM}} = \left((\boldsymbol{\Phi}\mathbf{K}\boldsymbol{\Phi}^\top)^{-1} + \boldsymbol{\Sigma}^{-1}\right)^{-1} \boldsymbol{\Sigma}^{-1} \mathbf{y}. \tag{47}$$

**Theorem E.1.** *With $\boldsymbol{\Phi}$ and $\mathbf{K}$ defined as above (Eqs. 44 and 45), we have $\hat{\boldsymbol{\mu}}^{\text{reg}} = \hat{\boldsymbol{\mu}}^{\text{SM}}$.*

In the proof we use a variant of Sherman–Morrison–Woodbury formula [Horn and Johnson, 2013, Eq. 0.7.4.1], specialized to symmetric positive definite matrices:

**Proposition E.1** (Symmetric SMW Formula). *Let $\mathbf{A} \in \mathbb{R}^{n \times n}$ and $\mathbf{R} \in \mathbb{R}^{m \times m}$ be symmetric positive definite matrices and let $\mathbf{X} \in \mathbb{R}^{n \times m}$. Then*

$$(\mathbf{A} + \mathbf{X}\mathbf{R}\mathbf{X}^\top)^{-1} = \mathbf{A}^{-1} - \mathbf{A}^{-1}\mathbf{X}(\mathbf{R}^{-1} + \mathbf{X}^\top \mathbf{A}^{-1}\mathbf{X})^{-1}\mathbf{X}^\top \mathbf{A}^{-1}.$$

*Proof of Theorem E.1.* It suffices to show that

$$\boldsymbol{\Phi}(\boldsymbol{\Phi}^\top \boldsymbol{\Sigma}^{-1} \boldsymbol{\Phi} + \mathbf{K}^{-1})^{-1} \boldsymbol{\Phi}^\top = \left((\boldsymbol{\Phi}\mathbf{K}\boldsymbol{\Phi}^\top)^{-1} + \boldsymbol{\Sigma}^{-1}\right)^{-1}.$$

We do this by direct calculation, using the symmetric SMW formula:

$$\boldsymbol{\Phi}(\boldsymbol{\Phi}^\top \boldsymbol{\Sigma}^{-1} \boldsymbol{\Phi} + \mathbf{K}^{-1})^{-1} \boldsymbol{\Phi}^\top = \boldsymbol{\Phi}\left[\mathbf{K} - \mathbf{K}\boldsymbol{\Phi}^\top(\boldsymbol{\Sigma} + \boldsymbol{\Phi}\mathbf{K}\boldsymbol{\Phi}^\top)^{-1}\boldsymbol{\Phi}\mathbf{K}\right]\boldsymbol{\Phi}^\top$$

$$= (\boldsymbol{\Phi}\mathbf{K}\boldsymbol{\Phi}^\top) - (\boldsymbol{\Phi}\mathbf{K}\boldsymbol{\Phi}^\top)\left((\boldsymbol{\Sigma} + (\boldsymbol{\Phi}\mathbf{K}\boldsymbol{\Phi}^\top))\right)^{-1}(\boldsymbol{\Phi}\mathbf{K}\boldsymbol{\Phi}^\top)$$

$$= \left((\boldsymbol{\Phi}\mathbf{K}\boldsymbol{\Phi}^\top)^{-1} + \boldsymbol{\Sigma}^{-1}\right)^{-1}.$$

The first equality follows by the SMW formula with $\mathbf{A} = \mathbf{K}^{-1}$, $\mathbf{R} = \boldsymbol{\Sigma}^{-1}$ and $\mathbf{X} = \boldsymbol{\Phi}^\top$, the second by multiplying out the terms, and the third by the SMW formula with $\mathbf{A}^{-1} = \boldsymbol{\Phi}\mathbf{K}\boldsymbol{\Phi}^\top$, $\mathbf{R}^{-1} = \boldsymbol{\Sigma}$ and $\mathbf{X}$ equal to the $d \times d$ identity matrix. $\square$

## E.2 Multi-task regression model

Consider multi-task setting with tasks indexed by $t = 1, \ldots, T$. We write $\mathcal{T} = [T]$ for the set of tasks. Multi-task setting can be reduced to single-task setting by viewing the task id as an additional sensitive attribute, and performing the same analysis as for the single-task setting, but with $\mathcal{G}' = \mathcal{T} \times \mathcal{G}$, with the dimension $d' = |\mathcal{G}'| = Td$.

Formally, we seek to fit $\boldsymbol{\mu}' \in \mathbb{R}^{|\mathcal{G}'|}$ based on the vector of empirical means $\mathbf{y}' \in \mathbb{R}^{|\mathcal{G}'|}$. The entries of $\boldsymbol{\mu}'$ and $\mathbf{y}'$ are denoted as $\mu'_{t,\mathbf{g}}$ and $y'_{t,\mathbf{g}}$ for the task $t$ and the intersectional group $\mathbf{g}$. We denote task-specific slices of these vectors as $\boldsymbol{\mu}'_t = \boldsymbol{\mu}'_{\{t\} \times \mathcal{G}}$ and $\mathbf{y}'_t = \mathbf{y}'_{\{t\} \times \mathcal{G}}$.

Features are indexed by elements of $\mathcal{J}' = \mathcal{S} \times \mathcal{J}$, where $\mathcal{S} = \mathcal{T} \cup \{\mathsf{glob}\}$, so there are task-specific and global features. The feature matrix $\mathbf{\Phi}' \in \mathbb{R}^{|\mathcal{G}'| \times |\mathcal{J}'|}$ uses the single-task feature matrix $\mathbf{\Phi} \in \mathbb{R}^{|\mathcal{G}| \times |\mathcal{J}|}$ (defined in Eq. 44) as a building block. The matrix $\mathbf{\Phi}'$ has a block structure, with $|\mathcal{T}| \times |\mathcal{S}|$ blocks of size $|\mathcal{G}| \times |\mathcal{J}|$ defined as

$$\mathbf{\Phi}'_{t,s} = \begin{cases} \mathbf{\Phi} & \text{if } s = t \text{ or } s = \mathsf{glob}, \\ \mathbf{0}_{|\mathcal{G}| \times |\mathcal{J}|} & \text{otherwise.} \end{cases}$$

We posit the regression model

$$\boldsymbol{\mu}' = \mathbf{\Phi}'\boldsymbol{\beta}',$$

where $\boldsymbol{\beta}' \in \mathbb{R}^{\mathcal{J}'}$. With the definition of $\mathbf{\Phi}'$ as above, this boils down to

$$\boldsymbol{\mu}'_t = \mathbf{\Phi}(\boldsymbol{\beta}'_t + \boldsymbol{\beta}'_{\mathsf{glob}}) \qquad \text{for all } t \in \mathcal{T}.$$

As before, we assume a Gaussian prior and Gaussian errors, $\boldsymbol{\beta}' \sim \mathcal{N}(\mathbf{0}, \mathbf{K}')$, $\mathbf{y}' \sim \mathcal{N}(\mathbf{\Phi}'\boldsymbol{\beta}', \mathbf{\Sigma}')$.

The error covariance $\mathbf{\Sigma}' \in \mathbb{R}^{|\mathcal{G}'| \times |\mathcal{G}'|}$ has a block-diagonal form with single-task error covariance matrices $\mathbf{\Sigma}_t \in \mathbb{R}^{|\mathcal{G}| \times |\mathcal{G}|}$, for $t \in [T]$, along the diagonal.

We consider a structured form of the prior covariance matrix $\mathbf{K}'$ specified by two vectors of hyperparameters: $\boldsymbol{\tau} = (\tau_A^2)_{A \subseteq [k]}$ and $\boldsymbol{\upsilon} = (\upsilon_A)_{A \subseteq [k]}$. The matrix $\mathbf{K}'$ is diagonal and positive definite, with entries

$$K'_{(s,A,\mathbf{c}),(s,A,\mathbf{c})} = \begin{cases} \tau_A^2 & \text{if } s \in \mathcal{T}, \\ \upsilon_A & \text{if } s = \mathsf{glob}. \end{cases}$$

It can also be viewed as a block-diagonal matrix with $|\mathcal{S}|$ diagonal blocks of size $|\mathcal{J}| \times |\mathcal{J}|$ defined as

$$\mathbf{K}'_{s,s} = \begin{cases} \mathbf{K} & \text{if } s \in \mathcal{T}, \\ \mathbf{V} & \text{if } s = \mathsf{glob}, \end{cases}$$

where $\mathbf{K}$ is the single-task prior matrix defined in Eq. (45), and $\mathbf{V} \in \mathbb{R}^{|\mathcal{J}| \times |\mathcal{J}|}$ is an analogous matrix based on the vector of hyperparameters $\boldsymbol{\upsilon}$ rather than $\boldsymbol{\tau}$, with diagonal entries

$$V_{(A,\mathbf{c}),(A,\mathbf{c})} = \upsilon_A. \tag{48}$$

The multi-task regression-based estimator is obtained by solving the resulting MAP regression problem. Similarly to the single-task case, the estimator takes form

$$\hat{\boldsymbol{\mu}}'^{\text{reg}} = \mathbf{\Phi}'\left(\mathbf{\Phi}'^{\top}\mathbf{\Sigma}'^{-1}\mathbf{\Phi}' + \mathbf{K}'^{-1}\right)^{-1}\mathbf{\Phi}'^{\top}\mathbf{\Sigma}'^{-1}\mathbf{y}', \tag{49}$$

with the individual task estimates denoted $\hat{\boldsymbol{\mu}}'^{\text{reg}}_t$.

The multi-task SureMap estimator, introduced in §3.2.2, takes form

$$\hat{\boldsymbol{\mu}}^{\text{SM}}_t = \mathbf{y}'_t + \left(\mathbf{\Lambda}^{-1} + \mathbf{\Sigma}_t^{-1}\right)^{-1}\mathbf{\Lambda}^{-1}\left(\hat{\boldsymbol{\theta}} - \mathbf{y}'_t\right), \tag{50}$$

where

$$\hat{\boldsymbol{\theta}} = \left(\mathbf{\Gamma}^{-1} + \sum_{t=1}^{T}(\mathbf{\Lambda} + \mathbf{\Sigma}_t)^{-1}\right)^{-1}\sum_{t=1}^{T}(\mathbf{\Lambda} + \mathbf{\Sigma}_t)^{-1}\mathbf{y}'_t, \tag{51}$$

and

$$\mathbf{\Lambda} = \mathbf{\Phi}\mathbf{K}\mathbf{\Phi}^{\top} \qquad \text{and} \qquad \mathbf{\Gamma} = \mathbf{\Phi}\mathbf{V}\mathbf{\Phi}^{\top}.$$

**Theorem E.2.** *With $\mathbf{\Phi}'$, $\mathbf{K}'$ and $\mathbf{\Sigma}'$ defined as above, we have $\hat{\boldsymbol{\mu}}'^{\text{reg}}_t = \hat{\boldsymbol{\mu}}^{\text{SM}}_t$ for all $t \in [T]$.*

In the proof we use the following identities:

**Proposition E.2.** *Let $\mathbf{A}, \mathbf{B} \in \mathbb{R}^{n \times n}$ be symmetric positive definite matrices and let $\mathbf{I}$ be the $n \times n$ identity matrix. Then*

$$(\mathbf{A}^{-1} + \mathbf{B}^{-1})^{-1}\mathbf{A}^{-1} = \mathbf{I} - \mathbf{A}(\mathbf{A} + \mathbf{B})^{-1} = \mathbf{B}(\mathbf{A} + \mathbf{B})^{-1}.$$

*Proof.* We have

$$(\mathbf{A}^{-1} + \mathbf{B}^{-1})^{-1}\mathbf{A}^{-1} = \left[\mathbf{A} - \mathbf{A}(\mathbf{A} + \mathbf{B})^{-1}\mathbf{A}\right]\mathbf{A}^{-1}$$
$$= \mathbf{I} - \mathbf{A}(\mathbf{A} + \mathbf{B})^{-1}$$
$$= (\mathbf{A} + \mathbf{B})(\mathbf{A} + \mathbf{B})^{-1} - \mathbf{A}(\mathbf{A} + \mathbf{B})^{-1}$$
$$= \mathbf{B}(\mathbf{A} + \mathbf{B})^{-1},$$

where the first equality follows by the SMW formula (Proposition E.1), and the remaining equalities follow by simple algebraic manipulations. □

*Proof of Theorem E.2.* Starting with Eq. (49), we have

$$\hat{\boldsymbol{\mu}}'^{\text{reg}} = \boldsymbol{\Phi}'\left(\boldsymbol{\Phi}'^{\top}\boldsymbol{\Sigma}'^{-1}\boldsymbol{\Phi}' + \mathbf{K}'^{-1}\right)^{-1}\boldsymbol{\Phi}'^{\top}\boldsymbol{\Sigma}'^{-1}\mathbf{y}'$$
$$= \left(\left(\boldsymbol{\Phi}'\mathbf{K}'\boldsymbol{\Phi}'^{\top}\right)^{-1} + \boldsymbol{\Sigma}'^{-1}\right)^{-1}\boldsymbol{\Sigma}'^{-1}\mathbf{y}'$$
$$= \left(\mathbf{I} - \boldsymbol{\Sigma}'\left(\boldsymbol{\Phi}'\mathbf{K}'\boldsymbol{\Phi}'^{\top} + \boldsymbol{\Sigma}'\right)^{-1}\right)\mathbf{y}', \tag{52}$$

where the second equality follows by the same reasoning as in the proof of Theorem E.1, and the third equality follows by Proposition E.2.

We next evaluate $\boldsymbol{\Phi}'\mathbf{K}'\boldsymbol{\Phi}'^{\top}$. Note that the matrices $\boldsymbol{\Phi}'$, $\mathbf{K}'$ and $\boldsymbol{\Sigma}'$ have the following block structure:

$$\boldsymbol{\Phi}' = \begin{pmatrix} \boldsymbol{\Phi} & & \boldsymbol{\Phi} \\ & \ddots & \vdots \\ & \boldsymbol{\Phi} & \boldsymbol{\Phi} \end{pmatrix}, \quad \mathbf{K}' = \begin{pmatrix} \mathbf{K} & & \\ & \ddots & \\ & & \mathbf{K} \\ & & & \mathbf{V} \end{pmatrix}, \quad \boldsymbol{\Sigma}' = \begin{pmatrix} \boldsymbol{\Sigma}_1 & & \\ & \ddots & \\ & & \boldsymbol{\Sigma}_T \end{pmatrix}.$$

Thus,

$$\boldsymbol{\Phi}'\mathbf{K}'\boldsymbol{\Phi}'^{\top} = \begin{pmatrix} \boldsymbol{\Phi} & & \\ & \ddots & \\ & & \boldsymbol{\Phi} \end{pmatrix}\begin{pmatrix} \mathbf{K} & & \\ & \ddots & \\ & & \mathbf{K} \end{pmatrix}\begin{pmatrix} \boldsymbol{\Phi}^{\top} & & \\ & \ddots & \\ & & \boldsymbol{\Phi}^{\top} \end{pmatrix} + \begin{pmatrix} \boldsymbol{\Phi} \\ \vdots \\ \boldsymbol{\Phi} \end{pmatrix}\mathbf{V}\left(\boldsymbol{\Phi}^{\top} \cdots \boldsymbol{\Phi}^{\top}\right)$$
$$= \begin{pmatrix} \boldsymbol{\Phi}\mathbf{K}\boldsymbol{\Phi}^{\top} & & \\ & \ddots & \\ & & \boldsymbol{\Phi}\mathbf{K}\boldsymbol{\Phi}^{\top} \end{pmatrix} + \begin{pmatrix} \mathbf{I} \\ \vdots \\ \mathbf{I} \end{pmatrix}\boldsymbol{\Phi}\mathbf{V}\boldsymbol{\Phi}^{\top}\left(\mathbf{I} \cdots \mathbf{I}\right)$$
$$= \begin{pmatrix} \boldsymbol{\Lambda} & & \\ & \ddots & \\ & & \boldsymbol{\Lambda} \end{pmatrix} + \begin{pmatrix} \mathbf{I} \\ \vdots \\ \mathbf{I} \end{pmatrix}\boldsymbol{\Gamma}\left(\mathbf{I} \cdots \mathbf{I}\right) = \boldsymbol{\Lambda}' + \mathbf{X}\boldsymbol{\Gamma}\mathbf{X}^{\top},$$

where in the last line we introduced the notation $\boldsymbol{\Lambda}'$ for the block-diagonal matrix with $T$ copies of matrix $\boldsymbol{\Lambda}$ along diagonal, and notation $\mathbf{X}$ for the matrix obtained by stacking $T$ copies of the $|\mathcal{G}| \times |\mathcal{G}|$ identity matrix $\mathbf{I}$ on top of each other. Plugging the last expression back into Eq. (52), we obtain

$$\hat{\boldsymbol{\mu}}'^{\text{reg}} = \left[\mathbf{I} - \boldsymbol{\Sigma}'\left(\boldsymbol{\Lambda}' + \mathbf{X}\boldsymbol{\Gamma}\mathbf{X}^{\top} + \boldsymbol{\Sigma}'\right)^{-1}\right]\mathbf{y}'$$
$$= \mathbf{y}' - \boldsymbol{\Sigma}'\Big[(\boldsymbol{\Lambda}' + \boldsymbol{\Sigma}')^{-1}$$
$$\qquad - (\boldsymbol{\Lambda}' + \boldsymbol{\Sigma}')^{-1}\mathbf{X}\left(\boldsymbol{\Gamma}^{-1} + \mathbf{X}^{\top}(\boldsymbol{\Lambda}' + \boldsymbol{\Sigma}')^{-1}\mathbf{X}\right)^{-1}\mathbf{X}^{\top}(\boldsymbol{\Lambda}' + \boldsymbol{\Sigma}')^{-1}\Big]\mathbf{y}', \tag{53}$$

where the second equality follows by the SMW formula (Proposition E.1) with $\mathbf{A} = \boldsymbol{\Lambda}' + \boldsymbol{\Sigma}'$, $\mathbf{R} = \boldsymbol{\Gamma}$, and $\mathbf{X} = \mathbf{X}$. We next focus on simplifying the last term in the bracket in Eq. (53).

Since $\boldsymbol{\Lambda}'$ and $\boldsymbol{\Sigma}'$ are block-diagonal, the matrix $(\boldsymbol{\Lambda}' + \boldsymbol{\Sigma}')^{-1}$ is also block-diagonal with blocks along the diagonal equal to $(\boldsymbol{\Lambda} + \boldsymbol{\Sigma}_t)^{-1}$ for $t = 1, \ldots, T$. Thus,

$$\mathbf{X}^{\top}(\boldsymbol{\Lambda}' + \boldsymbol{\Sigma}')^{-1}\mathbf{X} = \left(\mathbf{I} \cdots \mathbf{I}\right)\begin{pmatrix} (\boldsymbol{\Lambda} + \boldsymbol{\Sigma}_1)^{-1} & & \\ & \ddots & \\ & & (\boldsymbol{\Lambda} + \boldsymbol{\Sigma}_T)^{-1} \end{pmatrix}\begin{pmatrix} \mathbf{I} \\ \vdots \\ \mathbf{I} \end{pmatrix} = \sum_{t=1}^{T}(\boldsymbol{\Lambda} + \boldsymbol{\Sigma}_t)^{-1},$$

and similarly,

$$\mathbf{X}^\top(\mathbf{\Lambda}' + \mathbf{\Sigma}')^{-1}\mathbf{y}' = \sum_{t=1}^{T}(\mathbf{\Lambda} + \mathbf{\Sigma}_t)^{-1}\mathbf{y}'_t.$$

Therefore,

$$\left(\mathbf{\Gamma}^{-1} + \mathbf{X}^\top(\mathbf{\Lambda}' + \mathbf{\Sigma}')^{-1}\mathbf{X}\right)^{-1}\mathbf{X}^\top(\mathbf{\Lambda}' + \mathbf{\Sigma}')^{-1}\mathbf{y}'$$

$$= \left(\mathbf{\Gamma}^{-1} + \sum_{t=1}^{T}(\mathbf{\Lambda} + \mathbf{\Sigma}_t)^{-1}\right)^{-1}\sum_{t=1}^{T}(\mathbf{\Lambda} + \mathbf{\Sigma}_t)^{-1}\mathbf{y}'_t = \hat{\boldsymbol{\theta}}.$$

Plugging this back in Eq. (53), we obtain

$$\hat{\boldsymbol{\mu}}'^{\text{reg}} = \mathbf{y}' - \mathbf{\Sigma}'\left[(\mathbf{\Lambda}' + \mathbf{\Sigma}')^{-1}\mathbf{y}' - (\mathbf{\Lambda}' + \mathbf{\Sigma}')^{-1}\mathbf{X}\hat{\boldsymbol{\theta}}\right] = \mathbf{y}' - \mathbf{\Sigma}'(\mathbf{\Lambda}' + \mathbf{\Sigma}')^{-1}\left[\mathbf{y}' - \begin{pmatrix}\mathbf{I}\\ \vdots \\ \mathbf{I}\end{pmatrix}\hat{\boldsymbol{\theta}}\right].$$

Using, again, the fact that matrices $\mathbf{\Lambda}'$ and $\mathbf{\Sigma}'$ are block-diagonal, the task-specific blocks of $\hat{\boldsymbol{\mu}}'^{\text{reg}}$ must be equal to

$$\hat{\boldsymbol{\mu}}'^{\text{reg}}_t = \mathbf{y}'_t - \mathbf{\Sigma}_t(\mathbf{\Lambda} + \mathbf{\Sigma}_t)^{-1}(\mathbf{y}'_t - \hat{\boldsymbol{\theta}})$$

$$= \mathbf{y}'_t - \left(\mathbf{\Lambda}^{-1} + \mathbf{\Sigma}_t^{-1}\right)^{-1}\mathbf{\Lambda}^{-1}(\mathbf{y}'_t - \hat{\boldsymbol{\theta}})$$

$$= \mathbf{y}'_t + \left(\mathbf{\Lambda}^{-1} + \mathbf{\Sigma}_t^{-1}\right)^{-1}\mathbf{\Lambda}^{-1}(\hat{\boldsymbol{\theta}} - \mathbf{y}'_t) = \hat{\boldsymbol{\mu}}^{\text{SM}}_t,$$

where the second equality follows by Proposition E.2. $\qquad\square$

## F  Resources

### F.1  Data and model resources

We make use of data / models with the following sources / licenses:

1. Strack et al. [2014]: CC Attribution License.
2. Weerts et al. [2023]: MIT License.
3. Ardila et al. [2020]: CC0 License.
4. Radford et al. [2023]: Apache-2.0 License.
5. `https://archive.ics.uci.edu/dataset/2/adult`: CC BY 4.0 License
6. `https://huggingface.co/JaaackXD/Llama-3-70B-GGUF`: Meta Llama 3 License

We use the third and fourth resources to create a dataset of Whisper model evaluations on Common Voice utterances, which result in the **Common Voice** and **CVC** tasks described in §4.2; we also use the last two resources to create a dataset of in-context evaluations of Llama 3 on the Adult dataset, which results in the **Adult** task described in §4.1. Both resources are released under a CC BY 4.0 License and are available at `https://github.com/mkhodak/SureMap`.

### F.2  Computational

By far the most computation was required to generate the Common Voice, CVC, and Adult tasks, which was done on a machine with two RTX-8000 GPUs and took about a week. As described above, the corresponding datasets are made publicly available and easy to re-use without any GPU access. Given these dataset, the main experiments were run on a 40-core machine and take a couple hours, with the vast majority of this time spent running the structured regression approach of Herlihy et al. [2024]; see Figure 7 for a summary of the costs associated with each method evaluated in this paper. Code for both generating the task data and reproducing the method evaluations is available at `https://github.com/mkhodak/SureMap`.

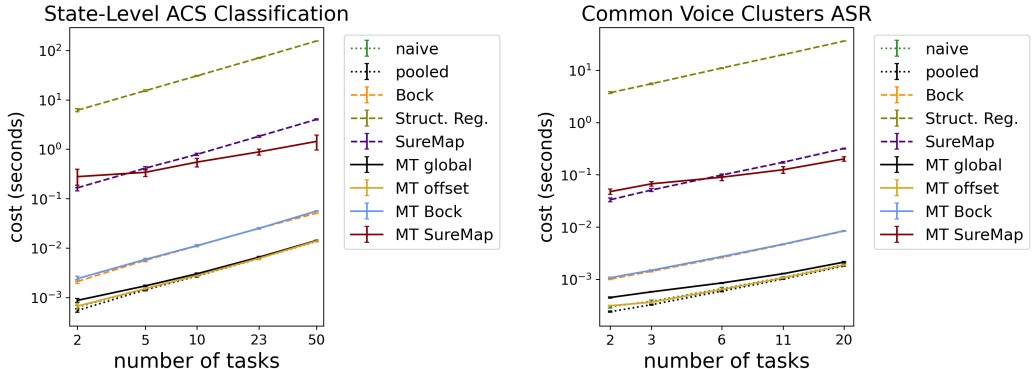

Figure 7: Cost of running the methods evaluated in this paper as a function of the number of tasks, with both axes scaled logarithmically. While they are 1-2 orders of magnitude more expensive than the baselines—which all have closed form expressions—SureMap and multi-task SureMap are also 1-2 orders of magnitude than structured regression [Herlihy et al., 2024]. Note that for the most part the runtime of all methods will usually be dwarfed by the cost of inference.

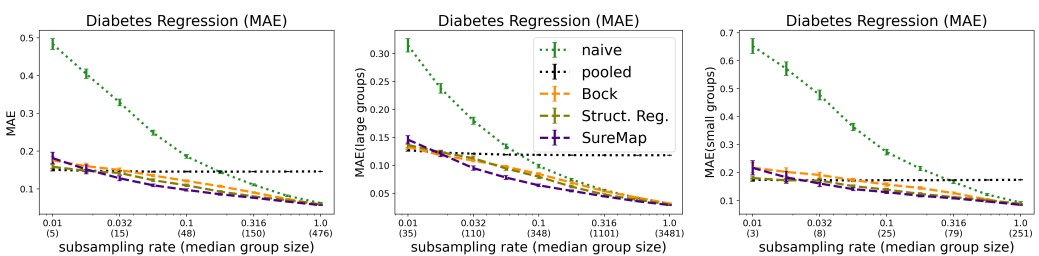

Figure 8: Evaluations on the regression variant of Diabetes, using MAE as the target metric, disaggregating by race, sex, and age. On the left the MAE is taken across all groups, while in the center it is only over large groups and on the left over small groups. Large and small are defined as the top and bottom half of all groups, respectively.

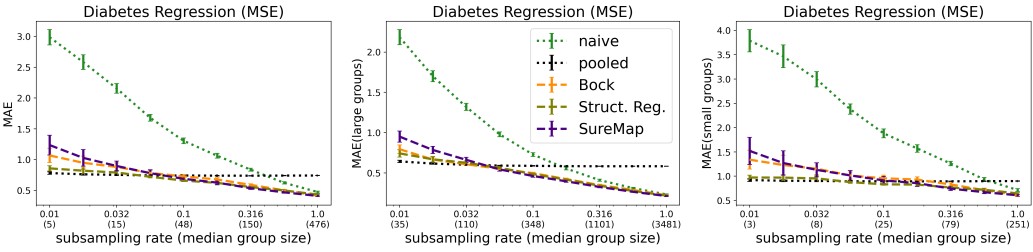

Figure 9: Evaluations on the regression variant of Diabetes, using MSE as the target metric, disaggregating by race, sex, and age. On the left the MAE is taken across all groups, while in the center it is only over large groups and on the left over small groups. Large and small are defined as the top and bottom half of all groups, respectively.

## G  Additional evaluations

In Figures 8 & 9, we compare evaluation methods on the regression variant of the Diabetes task, where the goal is to predict a patient's length of stay using ridge regression. In Figures 10 & 12, we compare evaluation methods on Diabetes and SLACS with AUC as the target metric. In Figures 11 & 13, we compare evaluation methods on Common Voice and CVC with the character error rate (CER) as the target metric. And finally, in Figures 14 & 15, we compare evaluation methods on Diabetes, Adult, Common Voice, SLACS, and CVC according to RMSE instead of MAE.

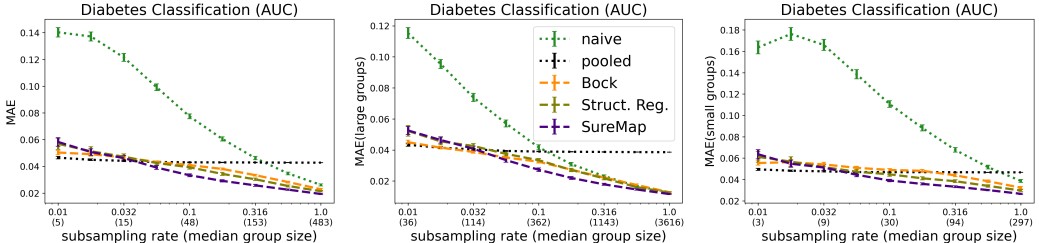

Figure 10: Single-task evaluations on the Diabetes classification setting (disaggregating by race, sex, and age) when using AUC as the target metric. In the left column the RMSE is taken across all groups, while in the center it is only over large groups and on the right over small groups. Large and small are defined as the top and bottom half of all groups, respectively.

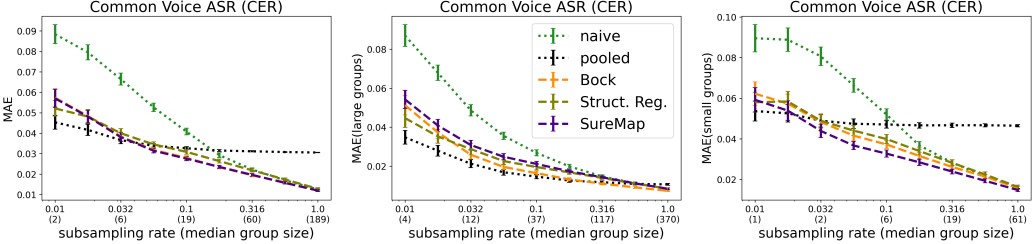

Figure 11: Single-task evaluation on the Common Voice ASR setting (bottom, disaggregating by sex and age) when using CER as the target metric. In the left column the MAE is taken across all groups, while in the center it is only over large groups and on the right over small groups. Large and small are defined as the top and bottom half of all groups, respectively.

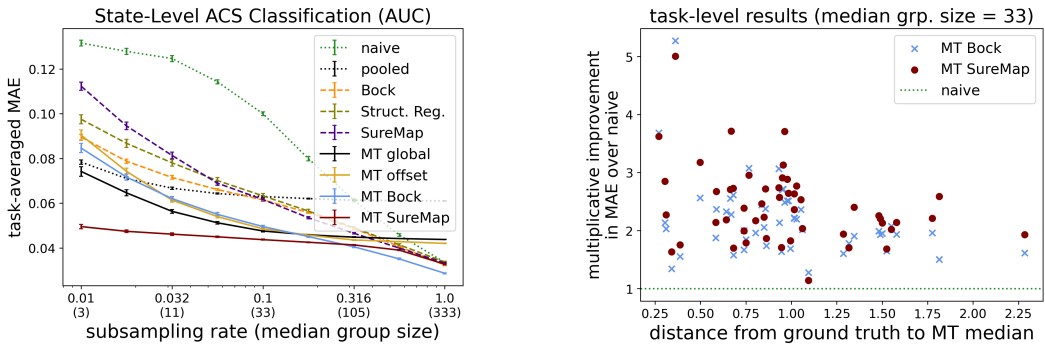

Figure 12: Multi-task evaluations on state-level ACS data (disaggregating by race, sex, and age) when using AUC as the target metric. On the left is the performance across different subsampling rates while on the right we show (multiplicative) performance improvement over the naive estimator on different tasks at subsampling rate 0.1.

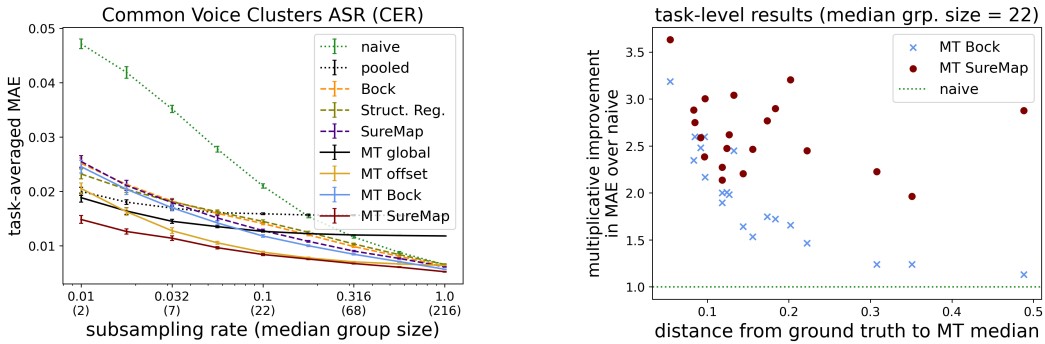

Figure 13: Multi-task evaluations on Common Voice clusters (disaggregating by sex and age) when using CER as the target metric. On the left is the performance across different subsampling rates while on the right we show (multiplicative) performance improvement over the naive estimator on different tasks at subsampling rate 0.1.

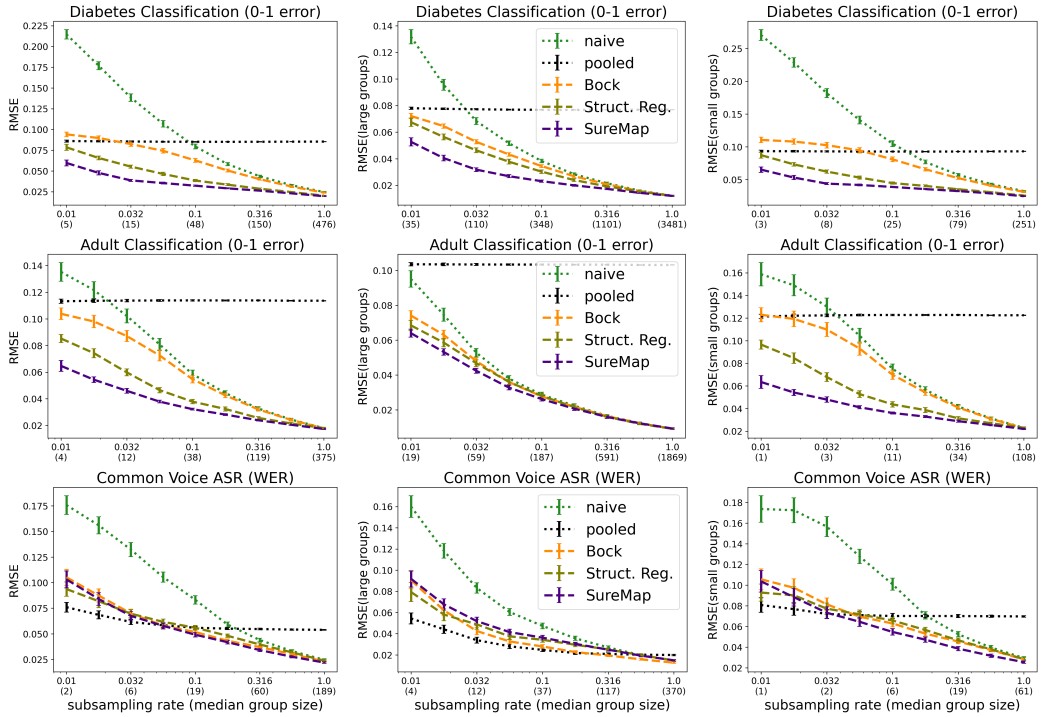

Figure 14: Single-task evaluations on the Diabetes classification setting (top, disaggregating by race, sex, and age), the Adult in-context classification setting (middle, disaggregating by race, sex, and age), and the Common Voice ASR setting (bottom, disaggregating by sex and age); these are the same evaluations as Figure 2 except with RMSE instead of MAE as the performance measure. In the left column the RMSE is taken across all groups, while in the center it is only over large groups and on the right over small groups. Large and small are defined as the top and bottom half of all groups, respectively.

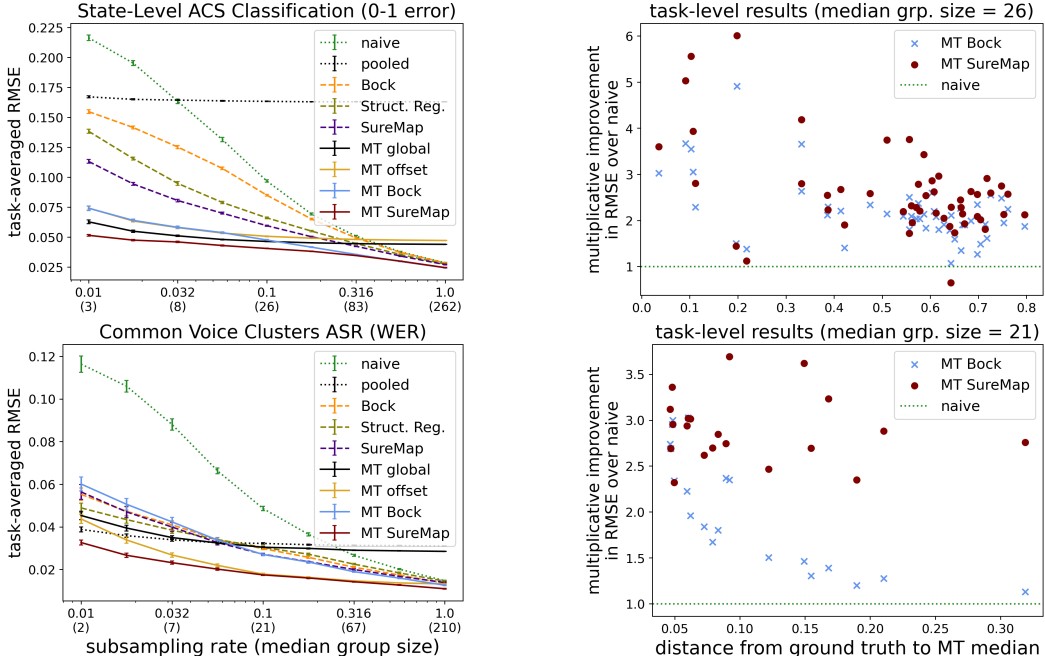

Figure 15: Multi-task evaluations on state-level ACS data (top, disaggregating by race, sex, and age) and Common Voice clusters (bottom, disaggregating by sex and age); these plots visualize the same evaluations as Figure 3 except they use RMSE instead of MAE as the performance measure. On the left is the performance across different subsampling rates while on the right we show (multiplicative) performance improvement over the naive estimator on different tasks at subsampling rate 0.1.

