# OpenReview forum: "SureMap: Simultaneous mean estimation for single-task and multi-task disaggregated evaluation"
_NeurIPS.cc/2024/Conference — NeurIPS 2024 poster_

### Official Review · Reviewer_UdGz · 2024-07-07

**Soundness:** 3
**Presentation:** 3
**Contribution:** 2
**Rating:** 5
**Confidence:** 2

**Summary:**

The paper proposed SureMap, a promising method for solving multi-task disaggregated evaluation problem. The key innovation of SureMap lies on transforming the problem into structured simultaneous Gaussian mean estimation and incorporating external data, e.g. from the AI system creator or from their other clients. Experiments on disaggregated evaluation tasks in multiple domains show the promising performance.

**Strengths:**

1) Well motivated.
2) Introduce several datasets for disaggregated evaluation and propose method that uses SURE to tune the parameters of a well-chosen
Gaussian prior before applying MAP estimation.
3) Experiments are promising. Competitive in both single task and multi-task evaluation.

**Weaknesses:**

As analyzed in Section 2.4:
1) Gaussian assumption may result in underperformance on heavy-tailed data.
2) Incorporating data from multiple clients may be costly.

**Questions:**

1) Can the authors provide some analysis or insights when the data contradicts with Gaussian assumption?
2) Can the authors visualize/show the cost when incorporating data from clients?

**Limitations:**

I don't think there are any negative societal impacts of this work.

---

> ### Author Rebuttal · Authors · 2024-08-07
>
> Thank you for your detailed review! We address your concerns and questions below:
>
> 1. [*Can the authors provide some analysis or insights when the data contradicts with Gaussian assumption?*]
> a. Please see our discussion of this issue in the general response (Issue 1: Assumption).
> 2. [*Can the authors visualize/show the cost when incorporating data from clients?*]
> a. Please see our discussion of efficiency in the general response (Issue 2: Computation) and in particular Rebuttal Figure 2. We will include this analysis and visualization in the revision.

---

> > ### Comment · Reviewer_UdGz · 2024-08-13
> > **Thanks for the rebuttal**
> >
> > Thanks for the rebuttal. I don't have further questions.

---

### Official Review · Reviewer_LQxL · 2024-07-10

**Soundness:** 3
**Presentation:** 3
**Contribution:** 3
**Rating:** 7
**Confidence:** 4

**Summary:**

This paper studies disaggregated evaluation, which aims to estimate the performance of models on various subpopulations. This problem is challenging due to small sample sizes in subpopulations, thus leading to inaccurate performance estimates. This issue is magnified when multiple clients use the same AI model and require individualized evaluations, which is referred to as multi-task disaggregated evaluation.

This paper designs a method that transforms the problem into a structured simultaneous Gaussian mean estimation problem. The method is comprised of two components: first, conduct maximum a posteriori (MAP) estimation for  Gaussian mean estimation and then apply cross-validation-free tuning using Stein’s unbiased risk estimate (SURE). Furthermore, the method employs an additive intersectional effect prior for capturing relationships between subpopulations with a limited number of hyperparameters.

The method, namely SureMap, is evaluated on various disaggregated evaluation tasks across different domains, including automated speech recognition (ASR) and tabular census data. The method shows high estimation accuracy for both single-task and multi-task settings over naive estimation, pooled estimation, and Bock estimation. Moreover, the method improves performance estimation even in data-poor regimes.

**Strengths:**

- This paper presents a novel method, SureMap, designed to improve the accuracy of performance estimation for models on various subpopulations. This is crucial for assessing the fairness and robustness of machine learning models, especially when subpopulations are small or data is scarce.
- Across tabular census data and automated speech recognition datasets in both single task and multitask settings, the proposed method outperforms previous estimation approaches by up to 50%. The results are consistent for various sampling rates and task numbers up to 50.

**Weaknesses:**

- The efficiency discussion of the method needs to be expanded. It would be better to list the complexity of the proposed method with existing methods and report the actual runtime.
- There are certain cases that the proposed method performs worse than the pooled estimation, such as in Figure 2. It would be better to analyze the reasons behind such results.

**Questions:**

- For subpopulations with few samples, how does this paper compute their ground truth and evaluate the estimation?
- It would be better to clarify the pooled estimator. For example, what does the notation $h$​ in Equation 3?
- It would be better to explain the working of the baselines, such as the Bock method.

**Limitations:**

This work has discussed its limitations in the setup of Gaussian distribution.

---

> ### Author Rebuttal · Authors · 2024-08-07
>
> Thank you for your detailed review! We address your concerns and questions below:
>
> 1. [*The efficiency discussion of the method needs to be expanded. It would be better to list the complexity of the proposed method with existing methods and report the actual runtime.*]
> a. Please see our discussion of efficiency in the general response (Issue 2: Computation). We will include this analysis in the revision.
> 2. [*There are certain cases that the proposed method performs worse than the pooled estimation, such as in Figure 2. It would be better to analyze the reasons behind such results.*]
> a. In Figure 2, pooling outperforms our approach at extremely low data regimes where the typical number of samples per group is 1. In this extreme regime, there may not be enough information to estimate group-level performance well and so the best thing to report is the overall mean (which is what the pooled estimator does).
> 3. [*For subpopulations with few samples, how does this paper compute their ground truth and evaluate the estimation?*]
> a. As noted in the first paragraph of Section 5, we exclude subpopulations with fewer than twenty samples, as we cannot obtain a reasonable ground truth for them.
> 4. [*It would be better to clarify the pooled estimator. For example, what does the notation $h$ in Equation 3?*]
> a. The pooled estimator just takes the average across all data samples, or equivalently a weighted average across the average performance on each subpopulation, with weights corresponding to the number of samples on each subpopulation. Equation 3 makes the latter explicit, and uses $h$ to index each subpopulation.
> 5. [*It would be better to explain the working of the baselines, such as the Bock method.*]
> a. The Bock estimator is described in Section 3.1, and specified explicitly in Equation 6. All other baselines are also specified in Equations 2, 3, and 5. We will make this clearer in the paper.

---

### Official Review · Reviewer_3ZUA · 2024-07-11

**Soundness:** 3
**Presentation:** 3
**Contribution:** 3
**Rating:** 5
**Confidence:** 2

**Summary:**

The author developed a disggregated evaluation method called SureMap, which has high estimation accuracy for both multi-task and single-task disggregated evaluations. SureMap transforms the problem into structured simultaneous Gaussian mean estimatio, incorporating external data. This method further combines Maximum A Posteriori (MAP) estimation and cross-validation-free tuning via Stein's risk estimate (SURE). Significant improvements in accuracy were observed in disggregated evaluation tasks.

**Strengths:**

1. The author introduces a new method, SureMap, which tunes the parameters of the selected Gaussian prior using SURE before applying MAP estimation. Only linear parameters are needed to recover several natural baselines for disggregated evaluation.
2.  The author introduces disggregated evaluation datasets for both single-task and multi-task settings.
3. SureMap shows good results in both single-task and multi-task settings.

**Weaknesses:**

1. I would like to know if there are any rules that need to be followed in the selection of disaggregated evaluation datasets to ensure fairness, as well as the reason why the disaggregated evaluation goal is set as the mean 0-1 error.
2. The fairness assessment is an application biased towards real-world scenarios. Does using Gaussian distribution as a prior align with real-world evaluations？

**Questions:**

Disaggregated evaluation is a core task in the fairness assessment of AI systems. This paper provides a possible solution and gives the corresponding analysis.

I would like to point out that my expertise does not directly align with the specific field of this article. Nevertheless, I have carefully read the paper several times, attempting to provide constructive feedback. I look forward to reading the insights of other reviewers, whose expertise is more closely related to the subject, to further inform my final score.

**Limitations:**

see the weaknesses and questions above

---

> ### Author Rebuttal · Authors · 2024-08-07
>
> Thank you for your detailed review! We address your concerns and questions below:
>
> 1. [*I would like to know if there are any rules that need to be followed in the selection of disaggregated evaluation datasets to ensure fairness, as well as the reason why the disaggregated evaluation goal is set as the mean 0-1 error.*]
> a. The goal of disaggregated evaluation is to evaluate fairness and thus help surface any fairness issues that need to be addressed. The question of selecting the right dataset and metric for disaggregated evaluation is an active area of research but beyond the scope of this paper. Some considerations include how well the data represents the intended uses of the AI system and how well the metric captures potential harms / benefits; see Barocas et al. (2021) for an in-depth discussion. In our case, for dataset selection we simply use all the available data, while for the performance metric note that 0-1 error is just one possible measure among many that our method can be applied to, including MAE, MSE, WER, AUC, and so on. In addition to the 0-1 error our experiments include results for MAE and WER.
> 2. [*The fairness assessment is an application biased towards real-world scenarios. Does using Gaussian distribution as a prior align with real-world evaluations?*]
> a. Please see our discussion of these points in the general response (Issue 1: Assumption). In particular, please note that we do not assume that the prior and individual-level distributions are Gaussian, only that the summary statistics are. As discussed in the response, this approximation is quite reasonable for numerous performance metrics of practical interest, including the 0-1 error, MSE, WER, AUC, and so on.
>
> ## References
> Barocas, Guo, Kamar, Krones, Morris, Wortman Vaughan, Wadsworth, Wallach. *Designing disaggregated evaluations of AI systems: Choices, considerations, and tradeoffs*. AIES 2021.

---

### Official Review · Reviewer_3rXK · 2024-07-11

**Soundness:** 3
**Presentation:** 3
**Contribution:** 3
**Rating:** 5
**Confidence:** 3

**Summary:**

This paper introduces SureMap, a new method for disaggregated evaluation of AI systems especially for multi-task setting. The authors model the problem as Gaussian mean estimation and use a structured covariance prior that captures intersectional effects. SureMap is evaluated on several datasets, including a new multi-task ASR dataset they introduce. They show SureMap generally outperforms existing methods, especially for small subgroups and in the multi-task setting.

**Strengths:**

1. This paper tackles an important problem in fairness and evaluation of AI systems and formulation as a Gaussian estimation problem with a structured prior.
2. Theoretical analysis showing SureMap can recover existing baselines.
3. This paper introduces the multi-task ASR datasets for evaluation.
4. The empirical results are strong, especially in the multi-task setting.

**Weaknesses:**

1. The Gaussian assumption may be too strong for some real-world settings
2. No comparison to some recent methods like GP-based approaches
3. The multi-task formulation assumes clients are willing to share data statistics

**Questions:**

1. Have you considered non-Gaussian models? For example, a t-distribution might better handle heavy-tailed performance data that can occur in practice.
2. The multi-task setting assumes clients are willing to share summary statistics. How realistic is this assumption? Could you explore privacy-preserving ways to share this information?
3. How does SureMap compare to GP-based approaches like in [1]? The method aims to handle low-data regimes well, which seems relevant here.

[1] Active assessment of prediction services as accuracy surface over attribute combinations. NeurIPS 2021

**Limitations:**

N.A

---

> ### Author Rebuttal · Authors · 2024-08-07
>
> Thank you for your detailed review! We address your concerns and questions below:
>
> 1. [*The Gaussian assumption may be too strong for some real-world settings [...] Have you considered non-Gaussian models? For example, a t-distribution might better handle heavy-tailed performance data that can occur in practice.*]
> a. Please see our discussion of these points in the general response (Issue 1: Assumption). In particular, as noted in the paper we view a derivation of a robust version of the SureMap objective using Student’s t-distribution as a good direction for future work.
> 2. [*The multi-task setting assumes clients are willing to share summary statistics. How realistic is this assumption? Could you explore privacy-preserving ways to share this information?*]
> a. There are many settings when this is not a concern, e.g. in situations when the companies are already sharing the performance statistics publicly as a form of disclosure to their customers or government regulators. In settings where this *is* a concern, it should be possible to apply techniques from the differential privacy (DP) literature to overcome these limitations (many DP techniques exist for releasing summary statistics, e.g. Biswas et al. (2020)). We leave a full investigation to future work.
> 3. [*How does SureMap compare to GP-based approaches like in [(Piratla et al., 2021)]? The method aims to handle low-data regimes well, which seems relevant here.*]
> a. Piratla et al. (2021) study a different low-data setting where they allow the user to *actively* sample points on which to evaluate performance. This difference makes it somewhat difficult to compare our methods.
>
> ## References
> Biswas, Dong, Kamath, Ullman. *CoinPress: Practical private mean and covariance estimation*. NeurIPS 2020.
> Piratla, Chakrabarti, Sarawagi. *Active assessment of prediction services as accuracy surface over attribute combinations*. NeurIPS 2021.

---

### Official Review · Reviewer_gnhf · 2024-07-12

**Soundness:** 3
**Presentation:** 3
**Contribution:** 2
**Rating:** 6
**Confidence:** 3

**Summary:**

This paper presents SureMap, a novel method for disaggregated evaluation, aimed at improving the estimation accuracy of performance metrics for machine learning models across different subpopulations. The proposed method is designed to address both single-task and multi-task settings, where multiple clients independently evaluate the same AI model on their respective data. SureMap leverages maximum a posteriori (MAP) estimation with a well-chosen Gaussian prior, fine-tuned using Stein’s unbiased risk estimate (SURE), to achieve high estimation accuracy. The authors evaluate SureMap across various domains, demonstrating significant improvements over existing baselines.

**Strengths:**

Originality: The introduction of SureMap for simultaneous mean estimation in disaggregated evaluation is novel, particularly in addressing both single-task and multi-task scenarios using a structured Gaussian prior and SURE for parameter tuning.
Quality: The paper is well-structured, with a thorough theoretical foundation, clear methodological development, and comprehensive experiments. The approach of combining MAP estimation with SURE tuning is well-justified and effectively demonstrated.
Clarity: The paper is clearly written, with detailed explanations of the methods and assumptions. The inclusion of theoretical proofs and detailed descriptions of the datasets and experiments adds to the clarity.
Significance: The ability to improve disaggregated evaluation accuracy has significant implications for fairness in AI, as it allows for better assessment of model performance across different demographic groups. This work has the potential to impact how AI systems are evaluated and deployed, ensuring more equitable outcomes.

**Weaknesses:**

Complexity of Implementation: While the method is theoretically sound, the practical implementation of SureMap may be complex, particularly the coordinate descent algorithm used for tuning parameters. This could pose challenges for practitioners.
Dependence on Gaussian Assumptions: The method relies on Gaussian assumptions for the prior and noise distributions. In real-world scenarios, data distributions may deviate significantly from Gaussian, potentially affecting the performance of SureMap.
Scalability: The scalability of the method in very large-scale settings is not fully explored. The computational cost associated with MAP estimation and SURE tuning might be prohibitive for very large datasets or a very high number of tasks.
Empirical Evaluation: While the empirical results are promising, they are primarily based on synthetic and semi-synthetic datasets. More extensive evaluation on real-world datasets across diverse domains would strengthen the findings.

**Questions:**

1.	Implementation Details: Can the authors provide more detailed pseudocode or a step-by-step guide for implementing the coordinate descent algorithm used in SureMap? This would help in better understanding and replicating the method.
	2.	Non-Gaussian Data: How robust is SureMap to deviations from the Gaussian assumptions? Have the authors considered alternative distributions, and how would the method need to be adapted for such cases?
	3.	Scalability: What are the computational requirements of SureMap for very large datasets or a high number of tasks? Can the authors provide any benchmarks or comparisons in terms of runtime and memory usage?
	4.	Real-World Applications: Can the authors provide examples of real-world applications where SureMap has been or could be successfully applied? This would help in contextualizing the method’s practical utility.

**Limitations:**

•	Gaussian Assumption: The reliance on Gaussian assumptions is acknowledged, and the authors suggest potential future work on other distributions like Student’s t.
	•	Data Integration Costs: The potential burden or cost of integrating data from multiple clients is mentioned, and the authors propose the use of model provider data as a mitigation strategy.
	•	Scalability: Although not deeply explored, the authors note the computational challenges and suggest future work to improve scalability and efficiency.
	•	Fairness and Over-Confidence: The authors caution against over-confidence in a model’s fairness based solely on disaggregated evaluations and emphasize the need for careful application of SureMap and similar methods.

---

> ### Author Rebuttal · Authors · 2024-08-07
>
> Thank you for your detailed review! We address your concerns and questions below:
>
> 1. [*While the method is theoretically sound, the practical implementation of SureMap may be complex, particularly the coordinate descent algorithm used for tuning parameters. This could pose challenges for practitioners. [...] Can the authors provide more detailed pseudocode or a step-by-step guide for implementing the coordinate descent algorithm used in SureMap?*]
> a. As discussed in the general response (Issue 2: Computation), we have updated the parameter optimization scheme post-submission to be L-BFGS. While SureMap is more involved than baselines such as using the naive estimate or the Bock estimator, it is as simple as or simpler than other recent approaches such as Structured Regression (Herlihy et al., 2024) and AAA (Piratla et al., 2021). The pseudo-code for our current method—which we will add in revision—can be summarized in two steps: (1) apply L-BFGS with SciPy’s default settings to the objective in Equation 10 (or Equation 11 for the multi-task case) and (2) run MAP estimation with the resulting mean and covariance.
> 2. [*The method relies on Gaussian assumptions for the prior and noise distributions. In real-world scenarios, data distributions may deviate significantly from Gaussian, potentially affecting the performance of SureMap. [...] How robust is SureMap to deviations from the Gaussian assumptions? Have the authors considered alternative distributions, and how would the method need to be adapted for such cases?*]
> a. Please see our discussion of these points in the general response (Issue 1: Assumption). In particular, please note that we do not assume that the prior and the noise are Gaussian; we only assume that the summary statistics are.
> 3. [*The scalability of the method in very large-scale settings is not fully explored. The computational cost associated with MAP estimation and SURE tuning might be prohibitive for very large datasets or a very high number of tasks. [...] What are the computational requirements of SureMap for very large datasets or a high number of tasks? Can the authors provide any benchmarks or comparisons in terms of runtime and memory usage?*]
> a. Please see our discussion of these points in the general response (Issue 2: Computation). In particular, SureMap is *not* computationally expensive for large datasets (it relies on cheap-to-compute summary statistics and so scales weakly with dataset size) nor for many tasks (as shown in Rebuttal Figure 2); in practice, we expect the cost of SureMap (less than a few seconds) is dominated by model inference costs. We will include this cost analysis in the revision.
> 4. [*While the empirical results are promising, they are primarily based on synthetic and semi-synthetic datasets. More extensive evaluation on real-world datasets across diverse domains would strengthen the findings. [...] Can the authors provide examples of real-world applications where SureMap has been or could be successfully applied? This would help in contextualizing the method’s practical utility.*]
> a. Please note that none of our benchmarks are fully synthetic, in the sense of being entirely generated by artificial distributions, and many do not involve any synthetic aspects at all. For example, our single-task ASR setting involves a widely used model (Whisper) evaluated on a real-world dataset (Common Voice). In terms of specific applications, the Diabetes task is motivated by a real-world use-case (Obermeyer et al., 2019) and has been used in previous disaggregated evaluation research (Miller et al., 2021; Herlihy et al., 2024). At a higher level, disaggregated evaluation is the central step in any fairness assessment (Barocas et al., 2021; Herlihy et al., 2024), so we expect SureMap to be of use in a broad range of real-world settings by AI companies, regulators, journalists, and researchers.
>
> ## References
> Barocas, Guo, Kamar, Krones, Morris, Wortman Vaughan, Wadsworth, Wallach. *Designing disaggregated evaluations of AI systems: Choices, considerations, and tradeoffs*. AIES 2021.
> Herlihy, Truong, Chouldechova, Dudík. *A structured regression approach for evaluating model performance across intersectional subgroups*. FAccT 2024.
> Miller, Gatys, Futoma, Fox. Model-based metrics: Sample-efficient estimates of predictive model subpopulation performance. ML4H 2021.
> Obermeyer, Powers, Vogeli, Mullainathan. *Dissecting racial bias in an algorithm used to manage the health of populations*. Science, 2019.
> Piratla, Chakrabarti, Sarawagi. *Active assessment of prediction services as accuracy surface over attribute combinations*. NeurIPS 2021.

---

### Author Rebuttal · Authors · 2024-08-07

First we would like to thank all the reviewers for their careful and thorough reviews. We are happy to see that reviewers found the problem we tackle important (Revs. gnhf, 3rXK), the paper well-structured and theoretically well-founded (Rev. gnhf), and the experiments convincing (Revs. LQxL, 3ZUA, 3rXK, UdGz). In this general response we would like to address two common points raised by reviewers about the assumptions and costs of our approach.

## Issue 1: Assumption
Several reviewers raised concerns about the assumptions used to derive our method (SureMap). The form of our estimator is motivated by a hierarchical model with a Gaussian prior and a Gaussian observation distribution. The Gaussian prior is used *only* to motivate the form of the estimator; the estimator is valid even when this assumption does not hold. What our approach does require is that the observations, which in our case correspond to summary statistics (e.g. average accuracy in each group), can be reasonably approximated by a Gaussian. Note that we do *not* assume that the individual-level performance metric (e.g. the accuracy of each example) is Gaussian.  We can usually expect the summary statistics (e.g., within-group average accuracy) to be approximately Gaussian due to the central limit theorem (CLT), which states that averages of samples from a bounded-variance distribution converge in distribution to a Gaussian with that distribution’s mean. In particular, this is guaranteed to be true for bounded performance measures, which includes many of the most important measures in ML including accuracy and (for the most part) word-error rate. CLT-like results also hold (and thus our method would also work) for other important performance measures that are not sample averages, including AUC. Thus **our Gaussianity assumption is well-justified in most target applications**.

A few reviewers correctly point out, as we also do in Section 2.4, that this approximation may work worse for heavier-tailed data. As a quick check of the severity of this problem, we re-evaluate SureMap and the baselines on Diabetes Regression (Figure 7 in the original submission) but using MSE rather than MAE as the performance measure to be estimated. MSE is likely to have heavier tails than MAE if the error distribution looks roughly Gaussian, since in that case the MSE follows a chi-squared distribution (sub-exponential) while MAE follows a one-sided Gaussian (sub-Gaussian). By comparing Rebuttal Figure 1 (left) with Submission Figure 7 (left), we see that on heavier tailed data SureMap no longer dominates, but it is also not significantly worse than the best baseline. This both suggests that SureMap is reasonably robust to data with heavier tails and reinforces our argument in Section 6 that disaggregated evaluation needs to be done with care. In this case for example, a statistician might first consider applying a variance-stabilizing transformation to the data before running their analysis.  For instance, in the case of MSE, it is known that the fourth root of a chi-squared random variable is approximately normally distributed (Hawkins et al., 1986), so one could apply SureMap to fourth root transformed MSE’s, and then transform back to produce estimates on the original scale. We view other approaches to making SureMap even more robust, e.g. by using Student’s t-distribution in the derivation of the optimization objective, as an important direction for future work.

## Issue 2: Computation
Some reviewers also raised concerns about the complexity of SureMap, including the computational complexity as the number of tasks increases and the implementation complexity of the coordinate descent scheme. While we also noted scalability as a limitation in Section 2.4 of the submission, after the deadline **we believe it is no longer a limitation**, largely for two reasons:
1. After the submission, we updated the optimization algorithm to use L-BFGS, which has a standard implementation in SciPy whose defaults work well in our settings. It is also quite efficient, with single and multi-task SureMap taking roughly one second to produce estimates for fifty tasks (c.f. Rebuttal Figure 2, noting the log scales), a very reasonable amount of time for running statistical data analysis. Notably, the same figure shows that multi-task SureMap scales slightly *better* than single-task SureMap with the number of tasks.
2. While the baselines are faster due to not having an optimization routine, we note that both they and SureMap are methods that evaluate model performance and therefore take as input the outputs of model inference. These are often quite expensive to generate in practice; for example, it took more than a GPU-day to generate the ASR dataset used in our evaluations. In comparison to this, a second or less of CPU time is a miniscule cost and thus not a concern for the practical usefulness of our method.

## References
Hawkins, Wixley. *A note on the transformation of chi-squared variables to normality*. The American Statistician, 1986.

---

### Decision · Program_Chairs · 2024-09-25

**Decision:**

Accept (poster)

**Comment:**

This paper studies estimating model performance in multiple distribution, a problem that has good relevance to fairness. The reviewers are in agreement that the theoretical analysis is adequate, the newly introduced datasets for evaluation are useful, and the experimental results are convincing. The rebuttal further addressed the reviewers' concerns about Gaussianity assumption and computational efficiency. Yet in the discussion phase there seems to be a lack of enthusiasm in championing this paper. Due to these considerations this paper is on the borderline.